# The potential of drone observations to improve air quality predictions by 4D-Var

**Hassnae Erraji**[1], **Philipp Franke**[1], **Astrid Lampert**[2], **Tobias Schuldt**[1], **Ralf Tillmann**[1], **Andreas Wahner**[1], and **Anne Caroline Lange**[1]

[1]Forschungszentrum Jülich GmbH, Institute of Climate and Energy Systems
– Troposphere (ICE-3), Jülich, Germany
[2]Institute of Flight Guidance, TU Braunschweig, Braunschweig, Germany

**Correspondence:** Anne Caroline Lange (ann.lange@fz-juelich.de)

**Abstract.** Vertical profiles of atmospheric pollutants, acquired by uncrewed aerial vehicles (UAVs, known as drones), represent a new type of observation that can help to fill the existing observation gap in the planetary boundary layer (PBL). This article presents the first study of assimilating air pollutant observations from drones to evaluate the impact on local air quality analysis. The study uses the high-resolution air quality model EURAD-IM (EURopean Air pollution Dispersion – Inverse Model), including the four-dimensional variational data assimilation system (4D-Var), to perform the assimilation of ozone ($O_3$) and nitrogen oxide (NO) vertical profiles. 4D-Var is an inverse modelling technique that allows for simultaneous adjustments of initial values and emissions rates. The drone data were collected during the MesSBAR (automated airborne measurement of air pollution levels in the near-earth atmosphere in urban areas) field campaign, which was conducted in Wesseling, Germany, on 22–23 September 2021. The results show that the 4D-Var assimilation of high-resolution drone measurements has a beneficial impact on the representation of regional air pollutants within the model. On both days, a significant improvement in the vertical distribution of $O_3$ and NO is noticed in the analysis compared to the reference simulation without data assimilation. Moreover, the validation of the analysis against independent observations shows an overall improvement in the bias, root mean square error, and correlation for $O_3$, NO, and $NO_2$ (nitrogen dioxide) ground concentrations at the measurement site as well as in the surrounding region. Furthermore, the assimilation allows for the deduction of emission correction factors in the area near the measurement site, which significantly contributes to the improvement in the analysis.

## 1 Introduction

In response to the increasing need for high-resolution and accurate air quality forecasts, extended efforts to improve the performance of chemical transport models (CTMs) have been made over recent decades. One of the effective means of improvement involves the use of advanced data assimilation techniques (Elbern et al., 2007; Liu et al., 2017; Klonecki et al., 2012). The aim is to combine observations and model data to obtain a better representation of the pollutants in the atmosphere as well as to optimise the input parameters, such as emissions, when considering inverse models. Although data assimilation holds significant potential for enhancing air quality modelling, its application is often still limited due to the scarcity of available observational data. In fact, the observational data types, which are usually used for assimilation (ground-based, airborne, and satellite observations), are certainly valuable for enhancing forecast accuracy, but they remain insufficient due to various constraints related to their availability, resolution, and especially their limited vertical coverage. Ground-based observations are the major source of information for regional CTMs and are generally taken from in situ monitoring networks. Even if they are fairly dense in the horizontal distribution on a regional scale, no information regarding the vertical distribu-

tion of air pollutants is provided. In contrast, lidar (light detection and ranging) remote sensing instruments and in situ sonde measurements can provide this information, but unfortunately, only a sparse and limited number of such stations exists. Similarly, ground-based Fourier transform infrared (FTIR) spectrometers, which are part of the Network for the Detection of Atmospheric Composition Change (NDACC), are capable of retrieving vertically resolved mixing ratios for a range of atmospheric constituents. However, the vertical resolution of these profiles is constrained by their dependence on a priori information, and the network's spatial coverage remains sparse (De Mazière et al., 2018; García et al., 2021). Multi-axis differential optical absorption spectroscopy (MAX-DOAS) is also capable of retrieving trace-gas and aerosol vertical profiles (Tirpitz et al., 2021). Airborne observations (e.g. In-service Aircraft for a Global Observing System – IAGOS – or flight campaigns) provide high-resolution vertical profiles during take-off and landing; however, the spatial coverage is still limited because of the high costs (Wang et al., 2022; Petetin et al., 2018; Tillmann et al., 2022). Satellite retrievals mainly provide the total column of air pollutants, thus providing little information on the vertical distribution of the air pollutant concentrations in the planetary boundary layer (PBL) and at the earth surface (Martin, 2008). Consequently, a significant observational gap exists in the PBL, which is the lowest part of the atmosphere characterised by the highest concentrations of air pollutants due to its vicinity to anthropogenic emission sources (Scheffe et al., 2009).

Uncrewed aerial vehicles (UAVs), also known as drones, are comparatively new measurement platforms that have begun to be widely utilised in recent years to obtain in situ measurements of atmospheric trace gases and aerosols within the lower atmosphere (Schuyler and Guzman, 2017; Yang et al., 2023), bringing many opportunities to improve air pollution monitoring. The increase in drone applications comes mainly from their numerous advantages, such as portability and flexibility, while being affordable. In addition, they can provide in situ observations of various atmospheric constituents with high temporal and vertical resolution (Lawrence and Balsley, 2013). However, drone measurements come with some limitations as, for instance, flights are complicated during strong wind conditions, require good visibility, and are often restricted to maximum altitudes due to aviation safety reasons. Nevertheless, they can fill the existing observational gap in the PBL and provide valuable information on the distribution of air pollutants.

Several studies present drone campaigns that have observed the atmospheric composition and meteorological parameters during the last 2 decades (Villa et al., 2016; Bretschneider et al., 2022). The measured data, mostly from the PBL region, were used for research on the atmospheric boundary layer (Wang et al., 2021) and pollutants' variability and distribution (Altstädter et al., 2015; Illingworth et al., 2014), as well as to study the properties of aerosols (Roberts

et al., 2008; Corrigan et al., 2008) and to qualify local emissions sources (Nathan et al., 2015). Furthermore, drone campaigns have been conducted in remote areas, such as the Arctic and Antarctic regions (Lampert et al., 2020), as well as during volcano eruptions (Diaz et al., 2012).

To our knowledge, the assimilation of drone observations has only been tested in the context of numerical weather prediction (NWP) models (Flagg et al., 2018; Leuenberger et al., 2020), and no study has yet explored their impact in the case of chemical data assimilation. Meteorological studies have shown that the assimilation of meteorological drone data has a positive impact on improving weather forecasts. This has prompted further ongoing research regarding the possibility of implementing drone observations in support of operational meteorology forecasting and for real-time data assimilation studies (O'Sullivan et al., 2021). Impact studies have revealed a large improvement in the vertical distribution of temperature, relative humidity, and wind, as well as a reduction in bias and root mean square error (RMSE), when drone observations are assimilated using a variational data assimilation system within high-resolution NWP models (Jonassen et al., 2012; Flagg et al., 2018; Jensen et al., 2021; Sun et al., 2020; Leuenberger et al., 2020).

Given the positive impact that has been reported in the case of meteorological applications, questions arise about the potential benefits and limitations of drone observations when assimilated within a CTM. In this study, the impact of drone data assimilation on air quality analyses is investigated using the regional and high-resolution EURopean Air pollution Dispersion – Inverse Model (EURAD-IM) with its four-dimensional variational (4D-Var) data assimilation system (Elbern et al., 2007). Vertical profiles of ozone ($O_3$) and nitrogen oxide (NO) collected during the MesSBAR CE1 (Automatisierte luftgestützte Messung der Schadstoffbelastung in der erdnahen Atmosphäre in urbanen Räumen – automated airborne measurement of air pollution levels in the near-earth atmosphere in urban areas) field campaign are assimilated. The potential of drone observations to improve air quality analysis and forecast is explored in a 2 d TS1 case study by applying the joint optimisation of initial values and emission rates. The aim is to investigate the ability of the 4D-Var system to adjust local emission rates using vertical profiles that were collected in a region characterised by diverse emission sources. This paper is structured as follows: in Sect. 2, the EURAD-IM and its 4D-Var data assimilation system are presented. The MesSBAR field campaign and the experimental design are described in Sect. 3. The results of the 4D-Var data assimilation experiments are discussed in Sect. 4. Finally, the summary and conclusions are given in Sect. 5.

## 2 The modelling system

### 2.1 The EURAD-IM model

EURAD-IM (EURopean Air pollution Dispersion – Inverse Model) is a three-dimensional high-resolution Eulerian CTM simulating air pollution in the troposphere at continental to regional scales. It has been used for several scientific studies for air quality forecasting, episode scenarios, data assimilation, and inverse modelling (Deroubaix et al., 2024; Gama et al., 2019; Elbern et al., 2007; Duarte et al., 2021; Franke et al., 2022, 2024). EURAD-IM is part of the regional Copernicus Atmosphere Monitoring Service (CAMS), providing daily air quality forecasts and reanalysis over Europe which enable continuous quality assurance using observations and inter-model evaluation (Marécal et al., 2015).

Table 1 presents a summary of the specific model settings and modules utilised in the EURAD-IM configuration employed in this study. EURAD-IM describes the transport by diffusion and advection of various trace-gas components emitted both by anthropogenic and biogenic sources and considers the gas-phase chemical transformation of about 110 chemical species with 265 reactions. The MADE (Modal Aerosol Dynamics model for Europe) module is employed to investigate aerosol dynamics within EURAD-IM, providing information on aerosol size distribution and chemical composition. This module simulates the formation and transformation of both primary and secondary aerosols, considering the interactions between the gas phase and aerosols. EURAD-IM accounts for the loss of chemical components through wet and dry deposition, as well as aerosol sedimentation. Moreover, EURAD-IM includes a 4D-Var assimilation system, as described in the subsequent section, along with the adjoint code derived from the forward code detailed in Elbern et al. (2007). The adjoint model incorporates the transport, diffusion, and gas transformation processes of the chemical species as well as secondary inorganic aerosol formation.

The CTM is driven by meteorological fields from the Weather Research and Forecasting (WRF) model (version 3.7; Skamarock et al., 2008) as thermodynamical forcing. The ECMWF (European Centre for Medium-Range Weather Forecasts) IFS (Integrated Forecasting System) global analysis (ERA5) is used for initialisation and boundary conditions for the WRF simulations. Chemical boundary conditions are generated by the CAMS global reanalysis data set (EAC4) that is produced by the ECMWF Composition Integrated Forecasting System (C-IFS). Anthropogenic emissions used for this study are provided by the German Environment Agency (Umweltbundesamt, UBA) for Germany and by the TNO-MACC_II inventory (Kuenen et al., 2014) for the rest of Europe. The emission data set is subject to processing in the EURAD Emission Module (EEM) (Memmesheimer et al., 1995) for seasonal and diurnal redistribution, as well as attributions to working days and weekends. The emission data are divided into point and area sources. The data contain emissions of gaseous air pollutants, i.e. carbon monoxide (CO), nitrogen oxides ($NO_x$), sulfur dioxide ($SO_2$), total non-methane volatile organic compounds (NMVOCs), and ammonia ($NH_3$), as well as the aerosols $PM_{10}$ (particulate matter with a diameter $< 10\,\mu m$) and $PM_{2.5}$ (particulate matter with a diameter $< 2.5\,\mu m$). Biogenic emissions are calculated online using the Model of Emissions of Gases and Aerosols from Nature (MEGAN), while wild-fire emissions are not considered here and did not play a role in the investigated case.

### 2.2 4D-Var data assimilation

The EURAD-IM data assimilation system is based on the 4D-Var method as described in Elbern and Schmidt (2001) and Elbern et al. (2007). The 4D-Var approach aims to determine the optimal model state by combining the prior information (e.g. provided by a forecast) with observational data over an assimilation window through the minimisation of the following cost function $\mathcal{J}$:

$$
\begin{aligned}
\mathcal{J}(\boldsymbol{x}_0, \boldsymbol{e}) &= \mathcal{J}_b(\boldsymbol{x}_0) + \mathcal{J}_0(\boldsymbol{x}_0) + \mathcal{J}_e(\boldsymbol{e}) \\
&= \frac{1}{2}\left(\boldsymbol{x}_0 - \boldsymbol{x}^b\right)^T \mathbf{B}^{-1}\left(\boldsymbol{x}_0 - \boldsymbol{x}^b\right) + \frac{1}{2} \\
&\quad \sum_{i=0}^{n}\left(\left(\boldsymbol{y}_i - \mathbf{H}_i\mathbf{M}_i\boldsymbol{x}_0\right)^T \mathbf{R}_i^{-1}\left(\boldsymbol{y}_i - \mathbf{H}_i\mathbf{M}_i\boldsymbol{x}_0\right)\right) \\
&\quad + \frac{1}{2}\left(\boldsymbol{e} - \boldsymbol{e}^b\right)^T \mathbf{K}^{-1}\left(\boldsymbol{e} - \boldsymbol{e}^b\right).
\end{aligned} \tag{1}
$$

Here, the optimisation is subject to the initial conditions $\boldsymbol{x}_0$ and the emission correction factor $\boldsymbol{e}$. The cost function equation includes an additional element (in contrast to the usual 4D-Var used for NWP) that accounts for emissions ($\mathcal{J}_e(\boldsymbol{e})$). The model state is mapped from the model space to the observation space by the observation operator $\mathbf{H}_i$ and the model operator $\mathbf{M}_i$, producing the model equivalents of each observation $\boldsymbol{y}_i$. The matrices $\mathbf{B}$, $\mathbf{R}$, and $\mathbf{K}$ represent the error covariance matrices associated with the a priori state vector $\boldsymbol{x}^b$, the observations $\boldsymbol{y}_i$, and a priori emissions $\boldsymbol{e}^b$, respectively. The matrix $\mathbf{R}$ considers only diagonal elements (i.e. it ignores any error correlation between different observations) while accounting for the uncertainties in the measurements and model representation error. The matrix $\mathbf{B}$ is estimated using error variances and the diffusion operator proposed by Weaver and Courtier (2001). Thus, $\mathbf{B}$ can be factorised as $\mathbf{B} = \mathbf{B}^{1/2}\mathbf{B}^{T/2}$ for use in the preconditioning of the highly underdetermined data assimilation system. The matrix $\mathbf{K}$ is defined as block diagonal, with non-zero entries for correlations between species and nearby emissions. The variance and correlation values are provided in Paschalidi (2015). The minimisation of the cost function $\mathcal{J}$ is performed through an iterative process using the quasi-Newton limited-memory L-BFGS algorithm (Liu and Nocedal, 1989), which includes

**Table 1.** Summary of EURAD-IM configuration.

|  | Processes | Modules and references |
|---|---|---|
| Transport | Advection | Walcek scheme (Walcek, 2000) |
| Gas-phase chemistry | Kinetic chemistry mechanism<br>Dry deposition<br>Wet deposition<br>Chemistry solver | RACM-MIM (Stockwell et al., 1997)<br>Zhang et al. (2003) scheme<br>Roselle and Binkowski (1999)<br>KPP (Sandu and Sander, 2006) |
| Aerosols | Aerosol dynamics<br>Secondary inorganic aerosols<br>Secondary organic aerosols | MADE (Ackermann et al., 1998)<br>HDMR (Rabitz and Aliş, 1999)<br>SORGAM (Schell et al., 2001) |
| Emissions | Biogenic emissions<br>Anthropogenic emissions | MEGAN (Guenther et al., 2012)<br>TNO–UBA emission inventory (Kuenen et al., 2014) |
| Assimilation | 4D-Var system<br>Minimisation algorithm<br>Background error covariance modelling | Elbern et al. (2007)<br>L-BFGS algorithm (Liu and Nocedal, 1989)<br>Weaver and Courtier (2001) |

the iterative integration of the forward and adjoint EURAD-IM.

## 3 The MesSBAR campaign analysis

### 3.1 Air quality measurements

The MesSBAR field campaign took place near Wesseling, Germany, on 22–23 September 2021. During these 2 d, a multicopter system composed of a drone and a set of low-cost air quality monitoring instruments was used to carry out vertical profile measurements of air pollutants during the morning hours. Among the instruments loaded on the multicopter, electrochemical sensors were used to monitor nitrogen oxide (NO), and a personal ozone monitor (POM) was deployed for assessing ozone ($O_3$) concentrations. The NO drone observations have an accuracy of 35 % at 40 ppbv with a precision of $\pm 2.5$ ppbv ($1\sigma$ at 30 s time resolution). POM provides an accuracy of 1.5 ppbv and a precision of 1.5 ppbv ($1\sigma$ at 10 s time resolution) in the observed $O_3$ mixing ratio range. The feasibility of using these sensors for measurements in the PBL was discussed in Schuldt et al. (2023) and Tillmann et al. (2022). A detailed description of the development, technical characteristics, and calibration of the multicopter system can be found in Bretschneider et al. (2022). The campaign's base was located within the proximity of the A555 highway, which is a much-frequented connection between the German cities of Cologne and Bonn. The measurements were conducted above agricultural land located about 1 km south of the town of Wesseling. The city centres of Cologne and Bonn are about 15 km north and 10 km south of the measurement location, respectively (Fig. 1). The Wesseling region is located within the Rhineland chemical region and is widely recognised as a leading chemical hub in Europe. Wesseling, in particular, hosts a re-

markable level of industrial activity attributed to the presence of major companies operating in the chemical and petroleum sectors (source: https://www.chemcologne.de/en/investments/the-rhineland-chemical-region, last access: 21 February 2024).

The objective of this campaign was to capture the early-morning evolution of air pollutant concentrations with the development of the PBL. Furthermore, the proximity to the highway allows for measurements of pollutants specifically originating from traffic sources.

The drone is operated by an autopilot system that uses an inertial navigation solution with an earth-related position based on GNSS (Global Navigation Satellite System) data. During the measurements, the autopilot controls a constant lateral position and a constant vertical climb rate of approximately $1\,\mathrm{m\,s^{-1}}$. Wind affects only the attitude of the copter, but given the low-wind situations during this campaign, the effect on the attitude can be neglected. The drone reached a maximum altitude of 350 m. This altitude limitation was imposed by air traffic restrictions in the area due to its proximity to the Cologne Bonn Airport. During each drone flight, two profiles were acquired: one ascending profile and one descending profile were done in a short period of time. For the assimilation experiments carried out with EURAD-IM, only the ascending profiles were utilised due to their higher accuracy (Schlerf et al., 2024). The measurements during the descending flights are strongly influenced by the turbulence generated by the drone's propellers, which reduces the data quality. In this study, the vertical profiles of $O_3$ and NO obtained from the multicopter are utilised and assimilated within EURAD-IM. The vertical resolution of these profiles is approximately 10 m, with 254 data points assimilated on 22 September 2021 and 257 on 23 September 2021 for both $O_3$ and NO. Additionally, observations from two ground-based stations situated on both sides of high-

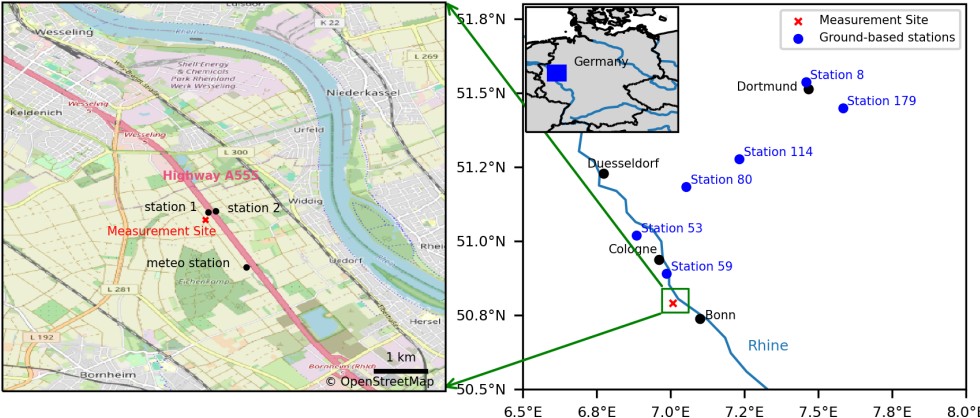

**Figure 1.** Geographic map displaying the MesSBAR measurement location, air quality ground stations, and meteorological station situated near the A555 highway. (Source: ©OpenStreetMap contributors 2023; distributed under the Open Data Commons Open Database License (ODbL) v1.0.)

way A555 (Fig. 1) are used to validate the simulation results. Furthermore, meteorological observations from an automatic weather station, located approximately 1 km southeast of the measurement site, are employed for comparing meteorological data, especially the wind fields.

## 3.2 Simulation set-up

The objective of this study is to investigate the impact of $O_3$ and NO drone profile assimilation on the air quality analysis using high-resolution EURAD-IM simulations. The model grid has a horizontal resolution of $5\,\mathrm{km} \times 5\,\mathrm{km}$ and is vertically divided into 30 layers defined by terrain following sigma coordinates between the surface and 100 hPa, with about 19 layers covering the lowest 1 km of the atmosphere. The EURAD-IM domain covers central Europe, including Germany with $271 \times 298$ grid points. The model output is adjusted to provide forecasts with a temporal resolution of 60 s, allowing for a more precise comparison with the high-resolution drone observations. To assess the impact of drone data assimilation on air quality forecasts, simulations are conducted both with and without data assimilation (Table 2). The joint initial value and emission rate optimisation mode of EURAD-IM is activated for this purpose. Two 24 h experiments are performed without assimilation: one on 22 September 2021 and the other on 23 September 2021. For these experiments, the model is initialised from a climatological chemical state with a spin-up simulation of 6 d (16–21 September 2021) prior to the campaign dates in order to establish a chemically balanced initial state. Moreover, two additional simulations focusing on $O_3$ and NO data assimilation are performed for 24 h on 22 and 23 September 2021. The assimilation window is deliberately selected to coincide with the availability of observations, aiming to minimise computational time in the simulations while also ensuring a meaningful lead time for emission optimisation. For drone

data assimilation, the observation error is considered as the sum of measurement and representativeness errors. The measurement error for $O_3$ is taken as the standard deviation of the measurements. For NO, the error $\epsilon_{\mathrm{meas}}$ is estimated according to Elbern et al. (2007) by defining a relative error $\epsilon_{\mathrm{rel}}$ and a minimal absolute error $\epsilon_{\mathrm{abs}}$:

$$\epsilon_{\mathrm{meas}} = \max(\epsilon_{\mathrm{abs}}, \epsilon_{\mathrm{rel}} \cdot y), \tag{2}$$

where $y$ is the individual observation. The absolute error used for NO is 2 ppbv, and the relative error is considered to be 20 % of the observed values.

The representation error is calculated by applying the corresponding formula from Elbern et al. (2007), which considers the grid cell spacing ($dx$), the representativeness length of the measurement location ($L_x$), and an absolute error specific to the measured species. The formula is expressed as TS2

$$\epsilon_{\mathrm{rep}} = \sqrt{\frac{dx}{L_x}} \times \epsilon_{\mathrm{abs}}. \tag{3}$$

The grid cell spacing ($dx$) corresponds to the spatial resolution of the measurement grid, while the representativeness length ($L_x$) indicates the effective range over which the measurement is considered representative. In this case study, $L_x$ is set to 3 km. The absolute error ($\epsilon_{\mathrm{abs}}$) varies by species: it is 2 ppbv for $O_3$ and 3 ppbv for NO. For the estimation of background errors, horizontal correlation lengths of 2.5, 10, and 20 km are employed at the surface, at the top of the PBL, and at the upper model levels, respectively.

## 3.3 Evaluation of the wind situation

The wind is a critical parameter that governs the dispersion of air pollutants and their transport, with a direct influence on emission optimisation within the framework of inverse CTMs. The wind conditions at the observation site are evaluated for two purposes: firstly to validate the suitability of the

**Table 2.** Model simulations presented in this study.

| Experiment name | Assimi­lation | Period | Assimilation window | Assimilated observations |
|---|---|---|---|---|
| REF_22SEP | No | 24 h, 22 September 2021 | – | – |
| REF_23SEP | No | 24 h, 23 September 2021 | – | – |
| DA_22SEP | Yes | 24 h, 22 September 2021 | 00:00–11:00 UTC | Six CE2 drone profiles of $O_3$ and NO |
| DA_23SEP | Yes | 24 h, 23 September 2021 | 00:00–09:00 UTC | Five drone profiles of $O_3$ and NO |

measurement site location for measuring local traffic emissions and secondly to assess the horizontal wind for applications to emission optimisation.

Figure 2a and b show CE5 the surface wind speed and direction observed by the nearby weather station during the flights' operation hours. The dominant wind direction is primarily from the southeast on 22 September 2021, with a maximum speed of $1.3\,\mathrm{m\,s^{-1}}$, while it comes from the south to southeast in the morning hours of 23 September 2021, with a maximum recorded speed of $2.0\,\mathrm{m\,s^{-1}}$. This indicates that the observation point is strategically located downwind of the nearest traffic emission source, which enabled the multicopter to successfully capture the emissions from the highway.

Apart from the surface conditions during the measuring period, each of the 2 d is characterised by a distinct wind situation, as shown in the horizontal wind profiles extracted from the WRF simulations in Fig. 2c and d. On 22 September 2021, the wind patterns exhibit vertical wind shear throughout the day and across all levels, changing direction from the east-southeast at lower altitudes to the west-northwest at higher altitudes. However, the wind intensity remains relatively low, measuring less than $3.0\,\mathrm{m\,s^{-1}}$. On 23 September 2021, the surface wind direction aligns with the observations during the campaign period. Nevertheless, at higher levels and beyond the campaign period, westerly and southwesterly winds dominate, and their speed increases with height. The maximum speed of $12.0\,\mathrm{m\,s^{-1}}$ is reached at 450 m between 05:00 and 07:00 UTC. The difference in the wind profiles between the 2 d may result in variations in the assimilation results, particularly with respect to emission optimisation.

## 4 Results

### 4.1 Impact on vertical profiles

In order to evaluate the impact of the drone data assimilation on the air pollutants' vertical distribution and given the lack of independent vertical profiles, the simulation results are first compared to the drone observations that are assimilated. Figure 3 presents the observed $O_3$ and NO drone profiles as well as vertical profiles resulting from the 4D-Var assimilation and the reference simulations. For both days, the 4D-Var analyses agree better with the drone observations in comparison to the reference forecast for both species, which indicates the successful assimilation of the drone observations. On 22 September 2021, an underestimation by the reference simulation is observed for the $O_3$ levels at altitudes above 200 m, with discrepancies reaching up to 15 ppbv, especially for the first three flights (F1, F2, and F3). The assimilation of drone profiles significantly reduces this underestimation. The bias was reduced by 98 % ($-4.58$ ppbv) for F1, 36 % ($-0.74$ ppbv) for F2, and 41 % ($-1.44$ ppbv) for F3, with an average reduction of 30 % ($-0.73$ ppbv) across all flights (Table 3). On 23 September 2021, the reference model run overestimates $O_3$ concentrations at both ground and near-surface levels. The most pronounced overestimations occur during the first three flights of the day (F7, F8, and F9), with discrepancies reaching up to 20 ppbv. Following the 4D-Var assimilation, the $O_3$ bias is reduced by more than 82 % ($-12.49$ ppbv) for F7, 56 % ($-2.86$ ppbv) for F8, and 25 % ($-0.96$ ppbv) for F9. As a result, the overall $O_3$ bias on the second day is reduced by approximately 55 % ($-3.46$ ppbv) (Table 3).

On both days, the reference simulations underestimate the NO vertical distribution at all heights, with the strongest discrepancies at ground level. Improvement due to the assimilation is accomplished mostly at surface and near-surface levels for the initial three flights of each day (F1, F2, F3, F7, F8, and F9), with more pronounced adjustments on the second day at ground level, while at higher levels during these same flights, the impact of the assimilation is minimal to non-existent, for instance, for flights F7 and F8 above 150 m. Overall, bias reductions of 24 % (6.78 ppbv), 33 % (11.61 ppbv), and 23 % (8.91 ppbv) were observed for F1, F2, and F3, respectively. On the second day, greater improvements were achieved, with reductions of 30 % (4.17 ppbv) for F7, 49 % (10.1 ppbv) for F8, and 57 % (15.29 ppbv) for F9. Because the pollutant concentrations are well-mixed in the PBL, a uniformly positive impact throughout the vertical can be seen in the NO analyses of the later flights of the day (F4, F5, F6, F10, and F11). The bias is reduced by 38 % ($-10.81$ ppbv) for F4, 54 % ($-15.26$ ppbv) for F5, and 49 % ($-14.66$ ppbv) for F6. On the following day, the bias reduction is smaller, with a 27 % ($-7.48$ ppbv) reduction for F10 and 18 % ($-5.58$ ppbv) for F11. Overall, the 4D-Var assimilation of drone observations leads to a substantial reduction in NO biases, with a 36 % reduction ($-11.34$ ppbv) on the first day and a 35 % reduction ($-8.52$ ppbv) on the second

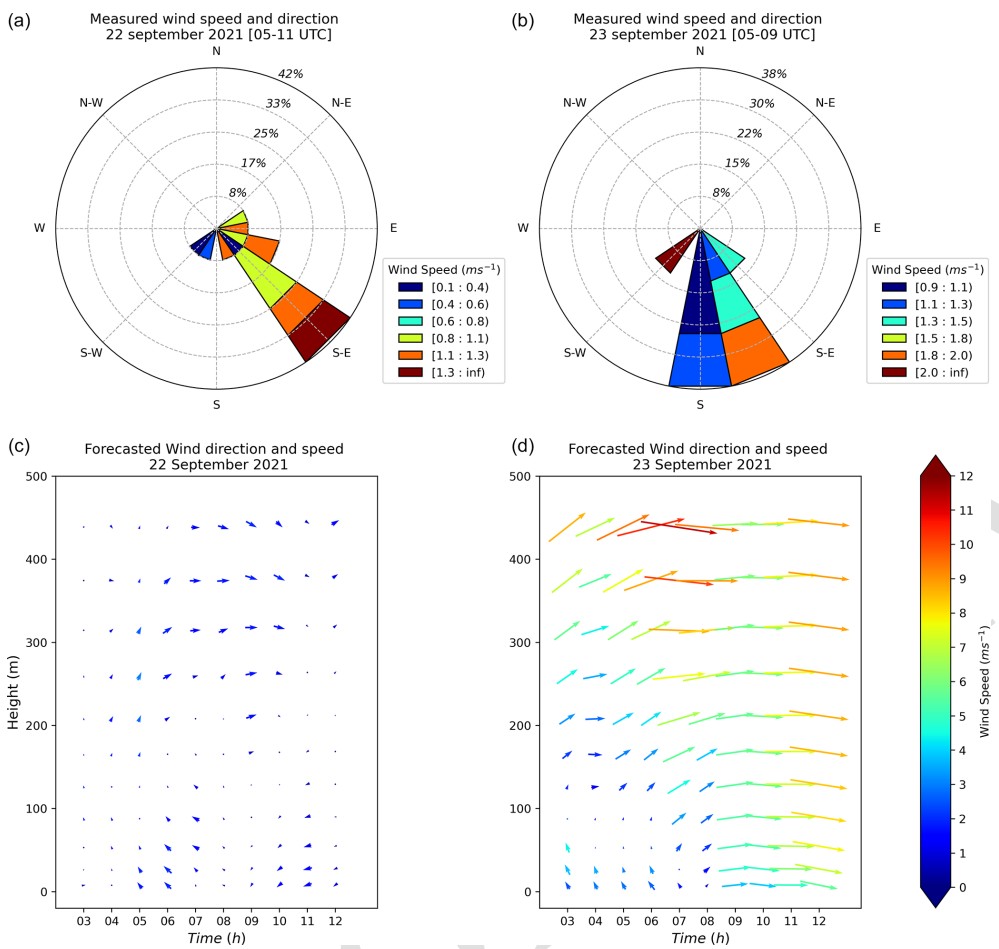

**Figure 2.** Observed surface wind speed and direction during the measurement period on 22 September 2021 **(a)** and 23 September 2021 **(b)**. Forecast of horizontal wind profiles for different hours for the lowest 500 m at the campaign location on 22 September 2021 **(c)** and 23 September 2021 **(d)**.

day, between the reference model forecast and observations (Table 3).

These results highlight the successful assimilation of drone observations by the EURAD-IM 4D-Var system. The accuracy of these findings is further examined and discussed in Sect. 4.3 through a validation process using independent observations.

## 4.2 Emission optimisation

The 4D-Var data assimilation method applied here aims at finding the best representation of the pollutants combining the knowledge provided by the EURAD-IM simulations and the drone $O_3$ and NO profile observations. The method relies on the assumption that the largest uncertainties in the modelled pollutant concentrations are based on uncertainties in initial values and emission rates. Emission correction factors for 25 anthropogenic pollutants can be deduced from the analysis. Consequently, it is worth looking at the emission factors being analysed to gain a first insight into the potential

to retrieve detailed information about emission assessment by applying this inverse modelling technique. However, their generalisation and significance should be carefully evaluated, mainly because of the limited number of drone profiles available, their unequal distribution during the course of the day, the resulting short assimilation windows, and the lack of a long-term statistical analysis.

The assimilation experiments performed with the $O_3$ and NO drone observations result in significant corrections of NO and $NO_2$ emission rates in the grids surrounding the observation site. The resulting emissions factors, which represent the ratio between the optimised emission rates and the input emission rates for each species, have variability that ranges from 1 to 4 for NO and from 1 to 6 for $NO_2$ in the DA_22SEP experiment. In contrast, the variability extends from 1 to 14 for both NO and $NO_2$ in the DA_23SEP experiment (Fig. A1). This indicates that an increase in emissions is analysed in the studied region. Figure 4 (first row) displays the original daily $NO_x$ emissions rates and the analysed emission changes on 22 and 23 September 2021. A

**Table 3.** $O_3$ and NO biases (model value minus observation; in ppbv) for each flight.

| Model runs | $O_3$ vertical profiles | | | | | | |
|---|---|---|---|---|---|---|---|
| | F 1 | F 2 | F 3 | F 4 | F 5 | F 6 | Daily absolute bias |
| REF_22SEP | −4.65 | −2.06 | −3.53 | −1.23 | −0.91 | −2.49 | 2.48 |
| DA_22SEP | 0.07 | −1.32 | −2.09 | −0.38 | −2.42 | −4.20 | 1.75 |
| | F 7 | F 8 | F 9 | F 10 | F 11 | | Daily absolute bias |
| REF_23SEP | 15.20 | 5.12 | 3.81 | 3.64 | 3.86 | | 6.33 |
| DA_23SEP | 2.71 | −2.26 | −2.85 | −3.92 | −2.63 | | 2.87 |
| | NO vertical profiles | | | | | | |
| | F 1 | F 2 | F 3 | F 4 | F 5 | F 6 | Daily absolute bias |
| REF_22SEP | −27.96 | −35.39 | −39.34 | −28.21 | −28.11 | −30.09 | 31.52 |
| DA_22SEP | −21.18 | −23.78 | −30.43 | −17.40 | −12.85 | −15.43 | 20.18 |
| | F 7 | F 8 | F 9 | F 10 | F 11 | | Daily absolute bias |
| REF_23SEP | −13.95 | −20.75 | −26.65 | −28.03 | −30.88 | | 24.05 |
| DA_23SEP | −9.78 | −10.65 | −11.37 | −20.55 | −25.30 | | 15.53 |

significant increase in $NO_x$ emissions is obtained in the DA_22SEP results, with changes in emission rates reaching up to $16\,Mg\,d^{-1}$ in the grid cells located north and northwest of the observation site. The emission of $16\,Mg\,d^{-1}$ represents approximately 3.46 % of the total daily $NO_x$ emissions in the analysed region, which is about $462\,Mg\,d^{-1}$. For DA_23SEP in contrast, the emission rates increase by up to $10\,Mg\,d^{-1}$ in the grid cells surrounding the observation site. Based on the chemical coupling with NO and $O_3$, carbon monoxide (CO), sulfur dioxide ($SO_2$), and sulfate ($SO_4$) emissions are optimised, resulting in emission correction factors between 1 and 3 (not shown).

To interpret the results and to investigate this discrepancy between the 2 d, the changes in $NO_x$ emissions are evaluated according to the emission source sectors. Figure 4 additionally shows the original $NO_x$ emissions and the analysed emission changes for three dominant polluter sectors in this region: power production, industry, and road transport. The original emission data set includes in total 12 GNFR (gridded nomenclature for reporting) sectors, while only these three sectors are substantially affected in the analysis. The DA_22SEP results indicate that 75 % of the emissions increase can be attributed to power generation and industrial activities. The remaining emission increase is mainly attributed to the road transportation sector. For the DA_23SEP results, almost half of the analysed emissions come from the road transport sector. In some grid cells, the additional road emissions of DA_23SEP are twice as high as those of DA_22SEP, reaching up to $6\,Mg\,d^{-1}$ compared to $1.5\,Mg\,d^{-1}$, respectively.

The area affected by the emission corrections differs for the 2 consecutive analysis days. This disparity lies in the different meteorological conditions, particularly in the variation in wind patterns, that occur during these days. As shown in Fig. 2 the prevailing winds in the studied region have low intensity and significant variability at the ground and high altitude on 22 September 2021, while on 23 September, the wind is more intense and predominantly originating from the west. This causes different dispersion situations for the pollutant during the 2 d.

This can be seen in Fig. 5, which shows tropospheric $NO_2$ columns observed by TROPOMI (Tropospheric Monitoring Instrument) on board the Sentinel-5 Precursor (Sentinel-5P) satellite. These data highlight that the accumulation of pollutants resulting in high $NO_2$ concentrations is very distinct for each individual day. On 22 September 2021, TROPOMI data show a highly polluted area north and northwest of the observation site, which does not persist on 23 September 2021. This might explain the increase in emissions rates seen in the DA_22SEP results to the north and northwest of the observation site. However, it is unfortunately not possible to directly obtain information about the $NO_2$ emissions from the TROPOMI data. Nevertheless, the 4D-Var assimilation algorithm seems to react to the high concentrations by attributing corrections to emission increases.

These results indicate the strong effects of the wind condition on the observability of the drone measurement. Nevertheless, it shows the potential that the drone observations have for emissions optimisation, especially for emissions that are emitted at higher altitudes, such as power plants and industries. Drawing definitive conclusions regarding the accuracy of emissions changes is consistently challenging, primarily due to the scarcity of emissions observations. Consequently, we will validate the 4D-Var analysis using independent ground-based observations, and we will analyse the contribution of emission changes to the observed improve-

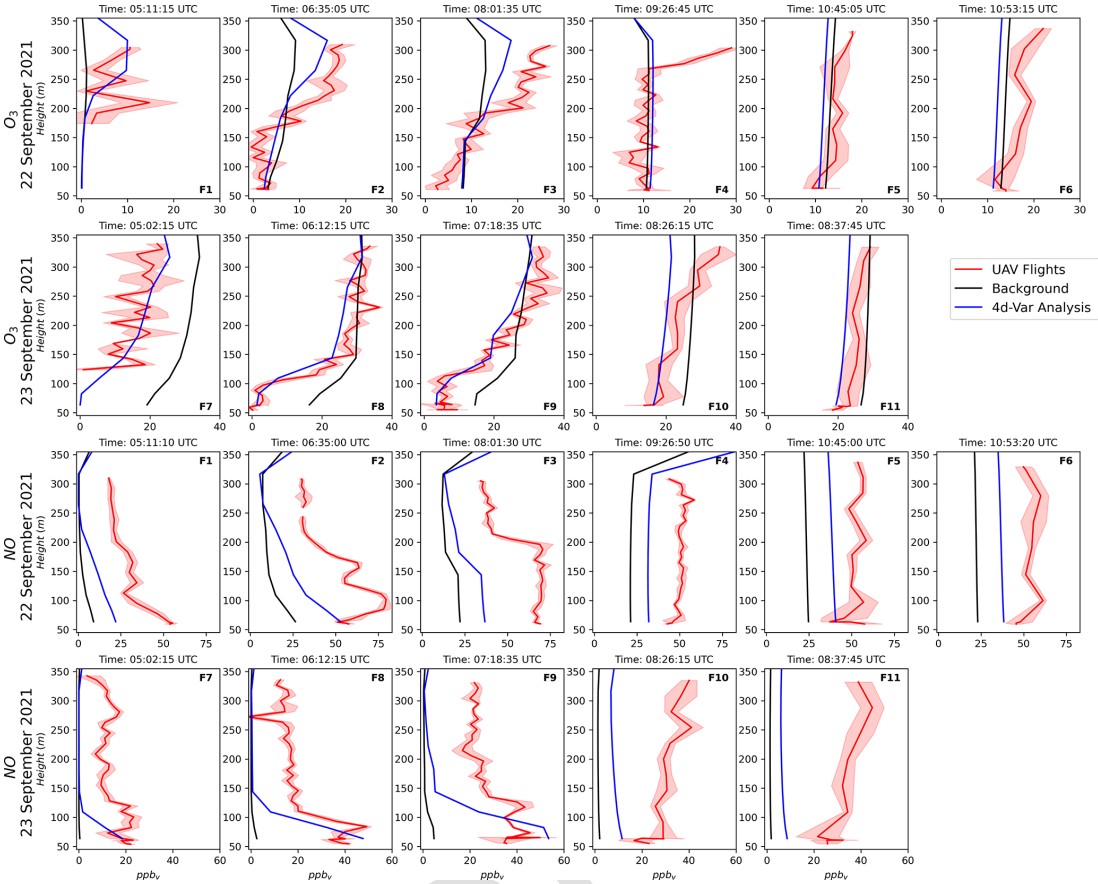

**Figure 3.** The vertical profiles of $O_3$ and NO measured by the drone system (red line) and compared to the 4D-Var analysis (blue line) and the reference run (black line) for all flights on 22–23 September 2021. The red shading highlights the standard deviation of the drone observations.

ments in order to evaluate the potential of drone observations in optimising emission rates.

## 4.3 Validation against independent observations

### 4.3.1 Local impact

To validate the impact of the drone data assimilation, we compare the experiment results with independent ground-based observations. Local observations from two monitoring stations located one on each side of the A555 highway but in the same grid cell as the assimilated data (Fig. 1) are used for this evaluation. Figure 6 shows the daily time series of observed $O_3$, NO, and $NO_2$ concentrations along with the modelled concentrations from both the reference and assimilation experiments. To evaluate the benefits of the drone data assimilation, the bias, RMSE (root mean square error), and Pearson correlation are examined for all experiments averaged over the assimilation window and over a 24 h period (Table 4) using the means of the observations from the two stations as reference.

The DA_22SEP experiment performance for the $O_3$ concentrations is almost similar to the reference experiment (REF_22SEP). Following the analysis of Sect. 4.1, this is expected because the a priori forecast and the drone observation for near-ground $O_3$ concentration agree well during this day. The main improvement during the first day is seen for the NO concentrations within the assimilation window as well as during the subsequent free forecast. The assimilation of drone observations results in a strong reduction in the bias of 87 % ($-20.48\,\mu g\,m^{-3}$) and the RMSE of 20 % ($-7.7\,\mu g\,m^{-3}$), with an amelioration in the Pearson correlation of 0.15 over the 24 h period. The daily $NO_2$ cycle is impacted by the assimilation due to its chemical coupling with $O_3$ and NO. Therefore, the assimilation experiment exhibits a better performance during the daytime relative to the reference experiment. However, during the late afternoon and nighttime, REF_22SEP performs better than DA_22SEP, as $NO_2$ is slightly overestimated. The best performance of the drone data assimilation results is obtained on 23 September 2021. A remarkable improvement in the $O_3$ concentration is noticed within the initial 7 h of the day, while a deterioration is observed between 16:00 and 24:00 UTC.

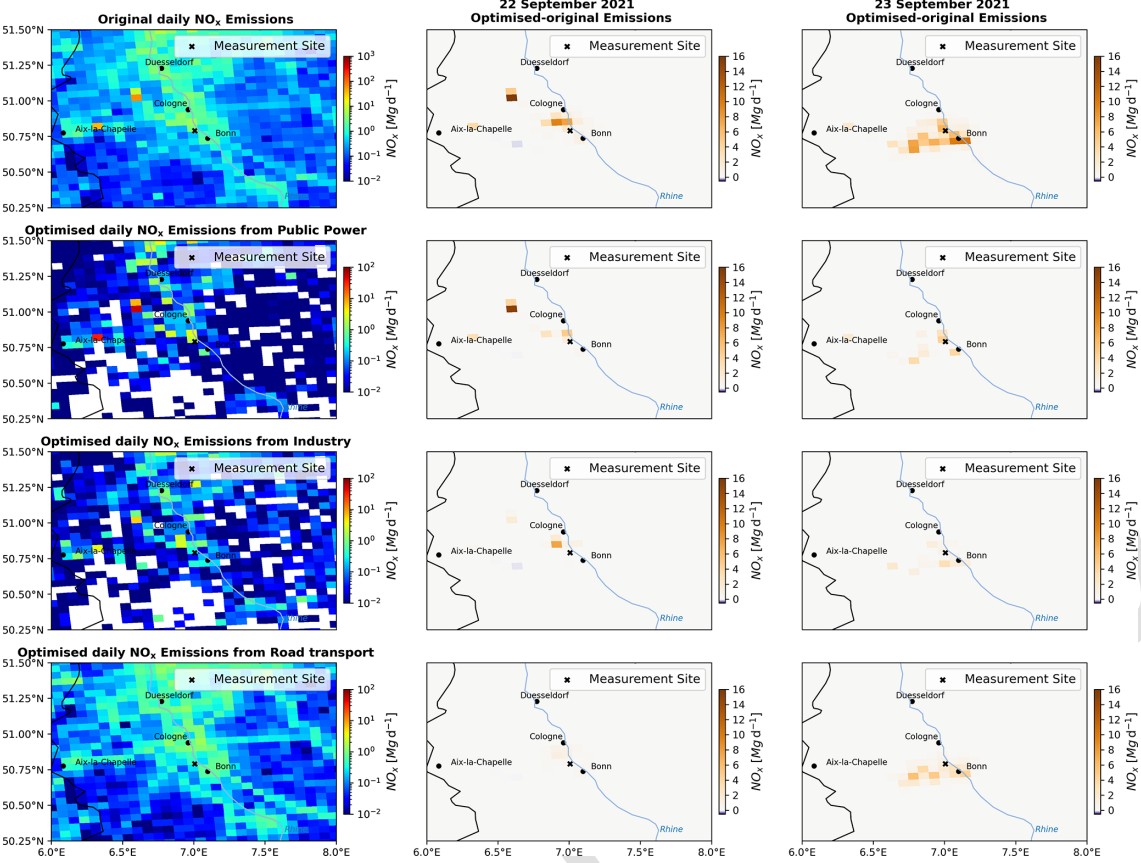

**Figure 4.** Daily $NO_x$ emissions within the analysed domain (left column) and the analysed $NO_x$ emission changes on 22 September (middle column) and 23 September (right column) 2021. The rows (from top to bottom) display the total $NO_x$ emissions and the emissions from public power production, industry, and road transport.

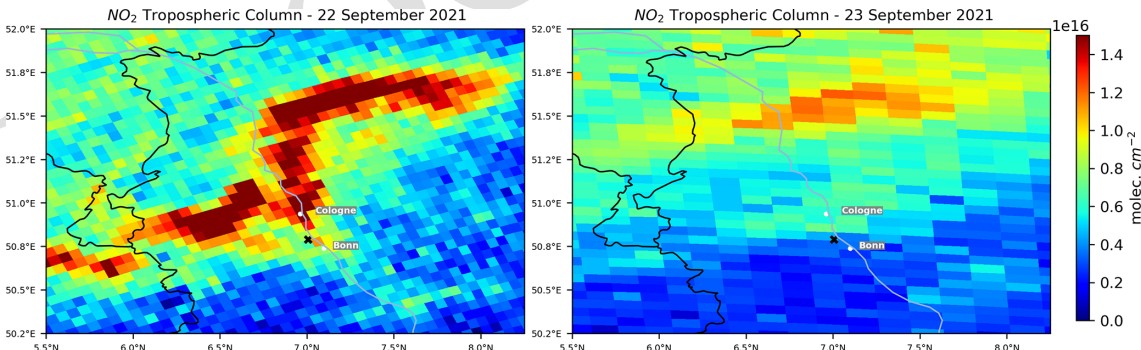

**Figure 5.** Maps of the TROPOMI $NO_2$ tropospheric columns (in molec. $cm^{-2}$) over the studied area on 22 September 2021 at 11:00 UTC (left) and on 23 September 2021 at 12:18 UTC (right). Source: https://browser.dataspace.copernicus.eu/ (last access: 30 May 2024).

The daily bias is reduced by 60 % ($-11.18\,\mu g\,m^{-3}$) and the RMSE by 46 % ($-11.06\,\mu g\,m^{-3}$), which also results in an improvement in the correlation of 0.22 during the assimilation window. An improvement in the assimilation results is achieved for NO concentrations. The assimilation experiment reduces the bias by 53 % ($-13.07\,\mu g\,m^{-3}$) and RMSE by 28 % ($-11.59\,\mu g\,m^{-3}$), with an amelioration in the cor-

relation of 0.5 over the 24 h evaluation period. For $NO_2$, a notable improvement can be seen in the forecast from DA_23SEP compared to REF_23SEP. Within the assimilation window, the bias reduced by 43 % ($-7.77\,\mu g\,m^{-3}$), the RMSE reduced by 29 % ($-6.68\,\mu g\,m^{-3}$), and the correlation improved by 0.19.

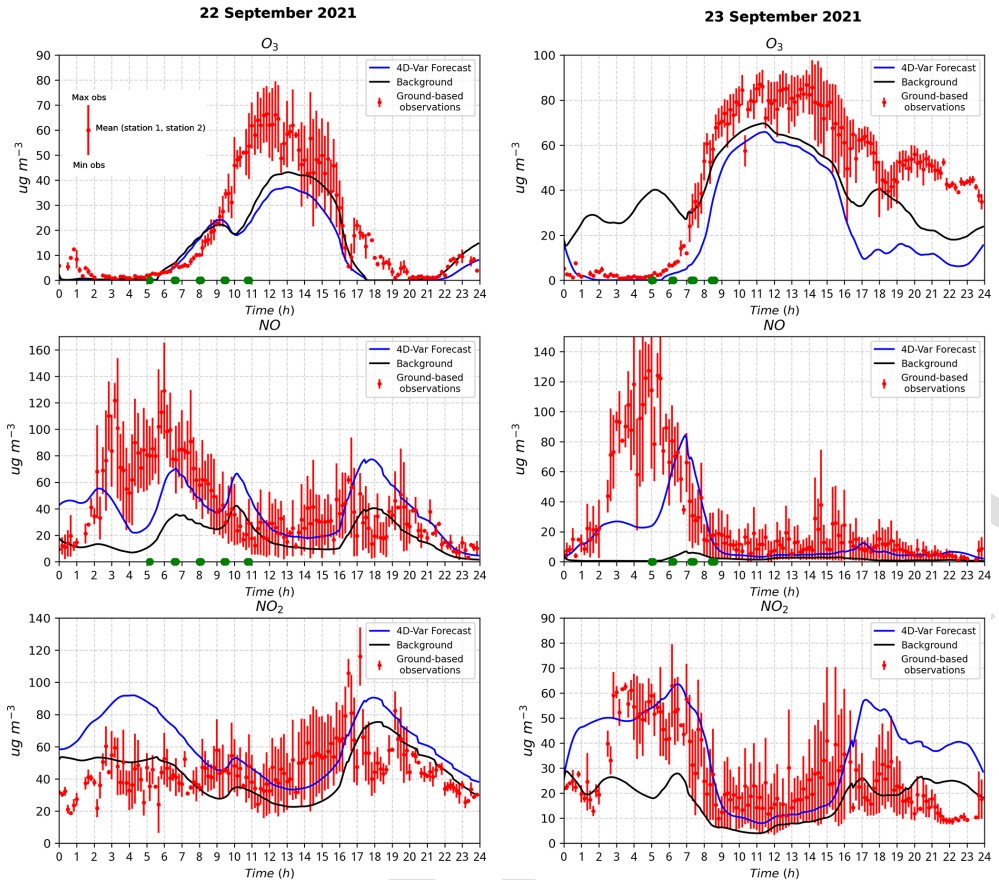

**Figure 6.** Temporal evolution of the $O_3$, NO, and $NO_2$ concentrations as observed by the ground stations (red line) and given by the model in the corresponding grid cell: the reference (black line) and the analysis (blue line) over the 24 h forecast period on 22 and 23 September 2021. Green dots highlight the time of the assimilated drone profiles.

**Table 4.** Statistical comparison of ground-based observations and model outputs (REF: reference run; DA: assimilation run) for $O_3$, NO, and $NO_2$ during the assimilation window and, in parentheses, the 24 h forecast on 22–23 September 2021. The bias and RMSE are in micrograms per cubic metre ($\mu g\,m^{-3}$).

| | | Statistics | $O_3$ | | NO | | $NO_2$ | |
|---|---|---|---|---|---|---|---|---|
| | | | REF | DA | REF | DA | REF | DA |
| 22 Sep | 2021 | Bias | −3.91 (−6.02) | −4.37 (−8.50) | −39.93 (−23.45) | −14.52 (−2.97) | 2.97 (−1.40) | 27.17 (15.73) |
| | | RMSE | 10.52 (11.42) | 10.93 (13.73) | 53.17 (37.84) | 38.44 (30.14) | 13.90 (17.66) | 32.08 (26.10) |
| | | Correlation | 0.83 (0.92) | 0.81 (0.92) | −0.14 (0.13) | −0.10 (0.28) | −0.13 (0.20) | 0.16 (0.18) |
| 23 Sep | 2021 | Bias | 18.53 (−5.37) | −7.35 (−21.60) | −52.62 (−24.82) | −23.61 (−11.75) | −17.83 (−9.45) | 10.06 (8.99) |
| | | RMSE | 24.10 (21.91) | 13.04 (26.32) | 66.16 (41.77) | 46.93 (30.18) | 22.84 (17.40) | 16.16 (18.70) |
| | | Correlation | 0.70 (0.71) | 0.92 (0.86) | −0.28 (−0.07) | 0.22 (0.56) | 0.40 (0.28) | 0.59 (0.49) |

These results indicate that the 4D-Var assimilation of the drone observations has the potential to improve the concentration of $O_3$, NO, and $NO_2$ during the early morning and daytime when optimising both the initial values and emissions rates simultaneously. The observed deterioration of the $O_3$ and $NO_2$ forecast during the late afternoon and nighttime in the DA_23SEP assimilation run is likely related to the $NO_x$ titration process. During the night, $O_3$ removal is

the dominant process in areas with significant NO emission sources (Sillman, 1999). Taking this into account may indicate that the drone data assimilation provides a higher estimate of $NO_2$ emissions during the night. Since the assimilation algorithm derives only one emission factor per day, the amplitude of the daily temporal emission profile is adjusted. It is assumed that the temporal emission profile is more certain than the emission strength. Deriving, for exam-

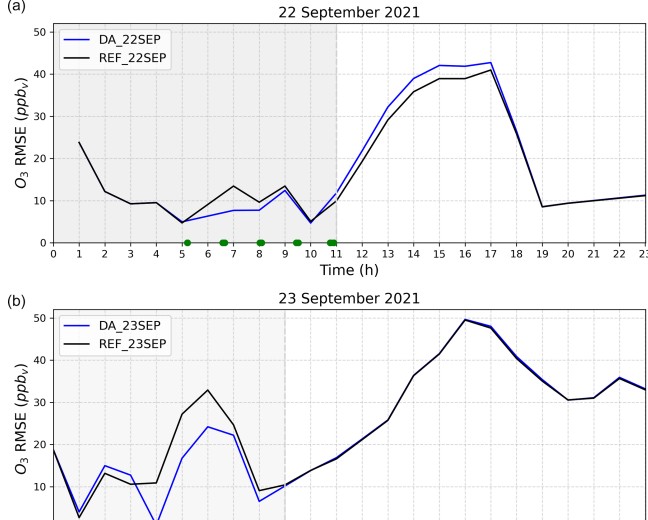

**Figure 7.** Temporal evolution of the RMSE (model − observations) (in ppbv) for $O_3$ calculated for the reference (black) and the data assimilation (blue) runs over the 24 h forecast period across all ground stations on 22 September 2021 **(a)** and 23 September 2021 **(b)**. Green dots highlight the time of the assimilated drone profiles.

ple, hourly emission factors instead would allow for more flexible adjustments of the emissions, which would be beneficial for the nowadays strongly regulated emission sources, such as power production (dependent on the availability of renewable energy). Previous studies demonstrated that the temporal distribution of traffic emissions significantly influences nighttime concentrations of $NO_2$ and $O_3$ (Menut et al., 2012). As the emission optimisation process maintains the same temporal variability, it is necessary to have 24 h data assimilation to improve the nighttime $O_3$ and $NO_2$ forecasts. Moreover, an inaccurately predicted PBL height can lead to uncertainties in the $O_3$ and $NO_2$ forecasts. A full analysis of the PBL representation is however beyond the scope of this study.

### 4.3.2 Regional impact

To further investigate the effect on a larger spatial scale, an additional validation is performed using independent ground-based observations from six different ground-based air quality monitoring stations situated in the vicinity of the observation site (Fig. 1, Table A1). For this validation, only stations that are impacted by the assimilation are selected. These are located at distances ranging from 12 to 85 km from the campaign location. Given the unavailability of NO observations, this validation considers only $O_3$ and $NO_2$. Although $NO_2$ is not assimilated in this study, it is indirectly influenced due to chemical coupling with the observed species and via the optimised $NO_x$ emissions. Figure 7 presents the hourly RMSE time series of $O_3$ concentrations for the assimilation and ref-

erence experiments, averaged over all selected stations. Corresponding results for $NO_2$ are depicted in Fig. A2. The individual RMSEs of $O_3$ and $NO_2$ within the assimilation window for all simulations per station are presented in Table 5.

Figure 7 shows that the $O_3$ RMSE for DA_22SEP and DA_23SEP is notably lower than that REF_22SEP within the data assimilation window. Outside the assimilation window, only a small added error is noted between 11:00 and 17:00 UTC for DA_22SEP, which appears similar to the results of the local validation, while no impact is observed during the subsequent free-forecast period for DA_23SEP. The largest RMSE reduction of 30 % takes place at Station 59 (−2.26 ppbv) on 22 September and of 40 % (−6.61 ppbv) on 23 September, as well as 35 % at Station 80 (−2.22 ppbv) on 22 September and 34 % (−4.98 ppbv) on 23 September. These stations are situated 12 and 43 km north of the campaign site, respectively. The smallest reductions occur at the stations of furthest distance, namely 5 % at Station 8 (−0.59 ppbv) on 22 September and 4 % (−0.46 ppbv) on 23 September and 2 % at Station 179 (−0.73 ppbv) on 22 September and 7 % (−1.22 ppbv) on 23 September, which are located approximately 85 km northeast of the campaign site. These results suggest that the positive impact of the drone data assimilation is transported to a broader area surrounding the campaign location, resulting in an improvement in $O_3$ concentrations across a larger area.

For $NO_2$, a significant RMSE reduction is found at Station 80, with a decrease of 72 % (−7.7 ppbv) for DA_22SEP. However, the RMSEs for Station 59 and Station 53 show an increase within the assimilation window. For DA_23SEP, better results can be seen for all stations except for the rural Station 59. The best reduction of 21 % is achieved at Station 80 (−4.16 ppbv) and 22 % at Station 114 (−2.80 ppbv).

Despite the simplicity of the current assimilation approach, which only incorporates data from a single grid box, a positive effect of assimilation is apparent even for stations situated at greater distances from the drone campaign location. This is attributed to the spatial spread of the analysis increment throughout large areas of the studied region.

### 4.4 Discussion of the potential and limitations of drone data assimilation

The analysis of the DA_22SEP and DA_23SEP experiments shows that the assimilation of drone observations has a positive impact on the vertical distribution of $O_3$ and NO and on the daily cycle of $O_3$ and $NO_x$ at ground level. These promising results underscore the significant potential of drone data assimilation in enhancing regional air quality analysis. Moreover, the assimilation process provides optimised emissions rates for each day. To investigate the role of emission optimisation in the analysis improvement, Table 6 presents the cost reduction for $O_3$ and NO, as well as the partial costs attributed to the optimisation of the initial values

**Table 5.** The $O_3$ and $NO_2$ RMSEs between observation data and model results obtained with (DA) and without (REF) drone data assimilation. The results are shown for every ground-based station for the assimilation window. The RMSE is in parts per billion by volume (ppbv).

| RMSE | | DA window | | DA window | |
|---|---|---|---|---|---|
| | | REF_22SEP | DA_22SEP | REF_23SEP | DA_23SEP |
| $O_3$ | Station 8 | 11.33 | 10.74 | 12.17 | 11.71 |
| | Station 53 | 10.29 | 9.66 | 8.19 | 7.29 |
| | Station 59 | 7.75 | 5.49 | 16.71 | 10.10 |
| | Station 80 | 6.35 | 4.13 | 14.58 | 9.60 |
| | Station 114 | 25.86 | 24.39 | 22.69 | 19.87 |
| | Station 179 | 27.96 | 27.23 | 17.55 | 16.33 |
| $NO_2$ | Station 8 | 18.11 | 17.49 | 24.05 | 22.92 |
| | Station 53 | 12.85 | 23.81 | 10.26 | 10.77 |
| | Station 59 | 24.25 | 44.34 | 16.88 | 24.45 |
| | Station 80 | 10.63 | 2.93 | 19.59 | 15.43 |
| | Station 114 | 24.14 | 25.82 | 12.81 | 10.01 |
| | Station 179 | 17.78 | 18.04 | 19.85 | 18.08 |

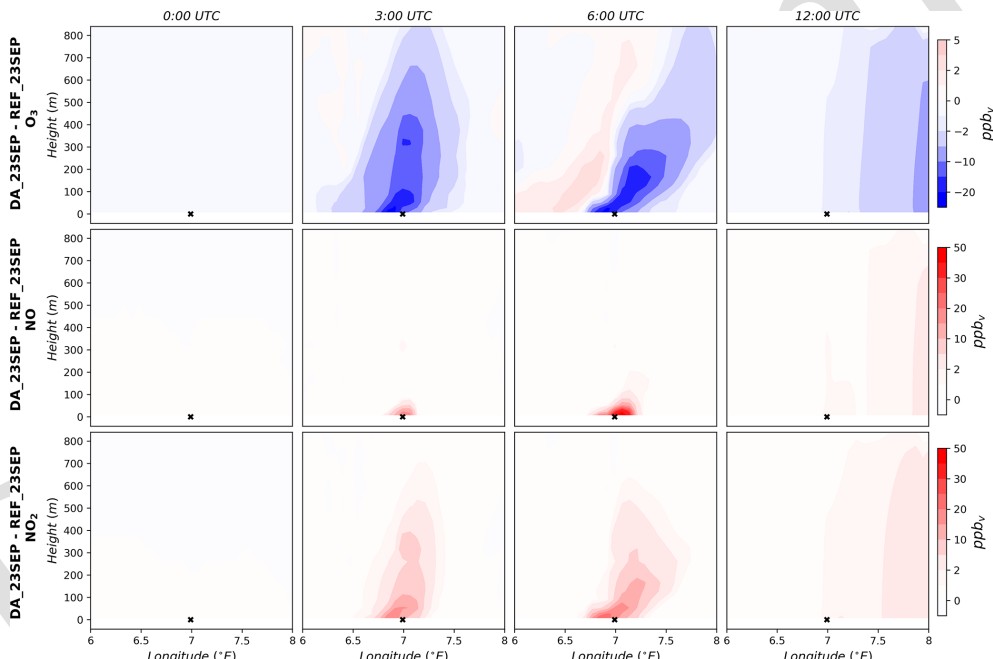

**Figure 8.** Vertical cross section of the analysis increment of $O_3$, NO, and $NO_2$ on 23 September 2021 at selected time steps. The cross section is located along the latitude of the MesSBAR campaign site.

(IVs) $\left(\frac{\mathcal{J}_b(x_0)}{\mathcal{J}(x_0,e)}\right)$ and the emission correction factors (EFs) $\left(\frac{\mathcal{J}_e(e)}{\mathcal{J}(x_0,e)}\right)$ TS3. For both assimilation experiments, the costs are reduced by more than 30 %, which confirms the successful assimilation of the drone profiles. In particular, the $O_3$ costs of DA_23SEP are highly reduced by 80 %, resulting in a precise alignment between the 4D-Var analysis and the $O_3$ observations. The partial costs vary between the 2 d. For DA_22SEP, the costs associated with IV are more than twice those of EF, which indicates important IV adjustments and a minimal impact of the emissions changes in the cost min-

imisation. In contrast for DA_23SEP, the effect of optimising the emissions is higher. This indicates that a significant part of the improvement observed in the analysis is due to the optimisation of EF. Therefore, the drone observations may also have significant potential for assessing local emissions. In a recent study by Wu et al. (2022), it was demonstrated that for high-altitude observations, the efficiency of emission rate optimisation is conditioned by favourable wind conditions and strong vertical diffusion.

Despite the observed improvements in the analysis, some limitations are noted. Firstly, the results reported in

**Table 6.** The percentage of cost reduction achieved for $O_3$ and NO, as well as the percentage of the partial costs attributed to initial value correction (IV) and emission correction factor (EF) relative to the total cost function.

|  | Cost reduction | | Partial costs | |
|---|---|---|---|---|
|  | $O_3$ | NO | EF | IV |
| DA_22SEP | 34 % | 41 % | 9 % | 25 % |
| DA_23SEP | 80 % | 36 % | 10 % | 4 % |

Sect. 4.1 show a limited impact on the NO vertical profiles on 23 September 2021. Although effective correction is achieved at the ground and near-ground levels, limited improvements are obtained for the NO concentrations at higher altitudes (above 150 m) for the first three profiles of the day. Figure 8 illustrates the vertically resolved analysis increment (4D-Var analysis – reference run) for $O_3$, NO, and $NO_2$ on 23 September 2021. A negative $O_3$ increment alongside a positive $NO_2$ increment is noted, both exhibiting a well-developed vertical spread. The NO increment is constrained near ground level during the early hours of the day. The reason behind this is the $NO_x$ titration process, where freshly emitted NO, including additional NO emissions resulting from emission optimisation, reacts with $O_3$ to produce $NO_2$. To achieve better results, a larger NO increment is needed. However, the NO observations from the drone exhibit high measurement errors compared to the background errors, which limits the effectiveness of assimilating these data.

Secondly, some suboptimal outcomes are observed in the free run, namely for $O_3$ and $NO_2$ ground concentrations, suggesting that the advantage of the drone data assimilation is limited to the assimilation window (Figs. 6, A3, and A4). Nevertheless, this result is not surprising and is completely explainable. Initially, it is important to note that the reference model simulation already provides underestimations of $O_3$ peaks during the afternoon and nighttime, which may be linked to uncertainties in the boundary layer height at night, vertical diffusion, and/or emission profiles. Through the 4D-Var assimilation of drone data, adjustments are made to the $NO_x$ emissions. However, in regions characterised by high $NO_x$ emissions, $O_3$ formation exhibits reduced sensitivity to $NO_x$ emissions but increased sensitivity to VOCs (Visser et al., 2019; Sillman, 1999). Thus, the inability to adjust $O_3$ concentrations and, consequently, $NO_2$ in our simulations is not a limitation specific to drone data assimilation.

## 5   Conclusion

In this study, drone profile measurements of $O_3$ and NO are assimilated using the 4D-Var data assimilation system of EURAD-IM. This represents the first application of drone data assimilation within a CTM. The primary objective is to assess the ability of drone observations to improve regional air quality analysis when the joint initial value and emission correction factor optimisation approach is applied. The research is conducted using data collected during the 2 d MesSBAR campaign in 2021. To evaluate the results, a comparison is made with ground-based observations obtained at stations very close to the drone flight base location. Moreover, regional validation is conducted using ground-based data from the European air quality monitoring network.

The 4D-Var assimilation of drone data has a positive impact on the representation of these pollutants in the PBL. First, significant improvements are noted in the $O_3$ and NO vertical profiles, with biases decreasing by 30 % and 55 %, respectively, on the first day and by 35 % on the second day for both species. Moreover, there is a noticeable impact on ground concentrations in the analysis. In the studied grid cell, biases are reduced by up to 60 % for $O_3$, 55 % for NO, and 43 % for $NO_2$ ground concentrations within the assimilation window. Furthermore, due to the pollution transport and the connected information propagation in the 4D-Var algorithm, a positive impact is seen in the ground concentrations of $O_3$ and $NO_2$ in locations farther from the measurement site. This study also identifies the assessment of emission correction factors as one component of the analysis improvements which underline the potential of the drone observations to be beneficial for emission optimisation.

There are some limitations to this study. Firstly, due to constraints in data availability, the study is restricted to assimilating drone data within a singular grid cell column. Therefore, it would be advantageous to include multiple measurement points distributed across the region, strategically positioned both upwind and downwind of emission sources. Another limitation of this study is the assimilation of data available only during a partial time window of the day. The inclusion of a more extensive observational data set covering longer periods, ideally over 24 h to enable an extended assimilation window, would greatly enhance the optimisation of emission rates.

In conclusion, the 4D-Var assimilation of drone data within the regional air quality model EURAD-IM yields promising results by improving the vertical distribution of pollutants and correcting ground concentrations. From a future perspective, a valuable extension of this work will be to conduct observing system simulation experiments (OSSEs) to evaluate the added value of integrating drone-based observations into the air quality forecasting system in comparison to conventional observations such as ground-based measurements and satellite data.

## Appendix A

**Table A1.** Information about the ground-based monitoring stations.

| Station number | Station code | Station name | Distance from campaign site | Station type | Latitude (°N) | Longitude (°E) | Altitude |
|---|---|---|---|---|---|---|---|
| 8 | DENW008 | Dortmund-Eving | 86.5 km | Suburban | 51.5369 | 7.4575 | 75 m |
| 53 | DENW053 | Köln-Chorweiler | 28.2 km | Suburban | 51.0193 | 6.8846 | 45 m |
| 59 | DENW059 | Köln-Rodenkirchen | 12.1 km | Rural | 50.8898 | 6.9852 | 45 m |
| 80 | DENW080 | Solingen-Wald | 43.2 km | Rural | 51.1838 | 7.0526 | 207 m |
| 114 | DENW114 | Wuppertal-Langerfeld | 56.8 km | Suburban | 51.2776 | 7.2319 | 186 m |
| 179 | DENW179 | Schwerte | 82.4 km | Suburban | 51.4488 | 7.5823 | 157 m |

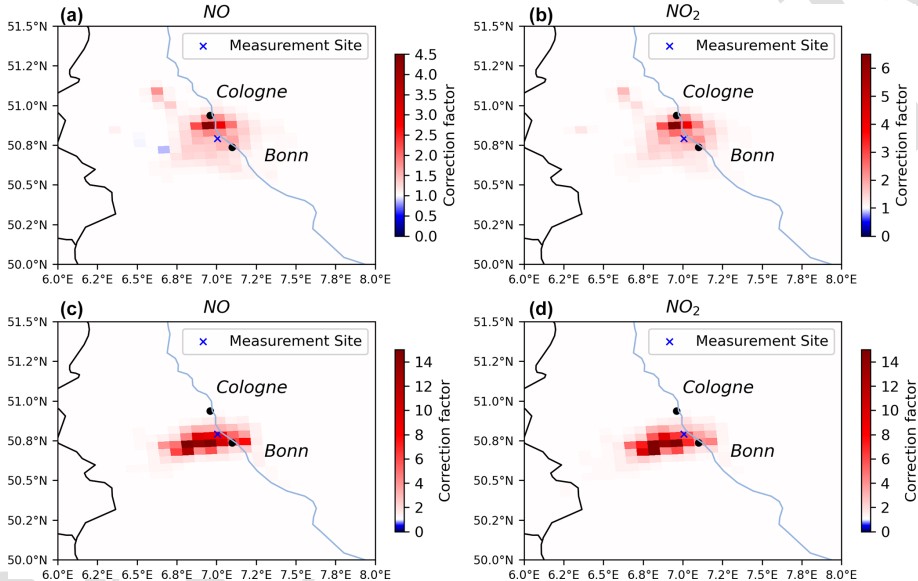

**Figure A1.** Emission correction factors of NO and $NO_2$ resulting from the conducted assimilation experiments on 22 September 2021 (**a** and **b**) and 23 September 2021 (**c** and **d**).

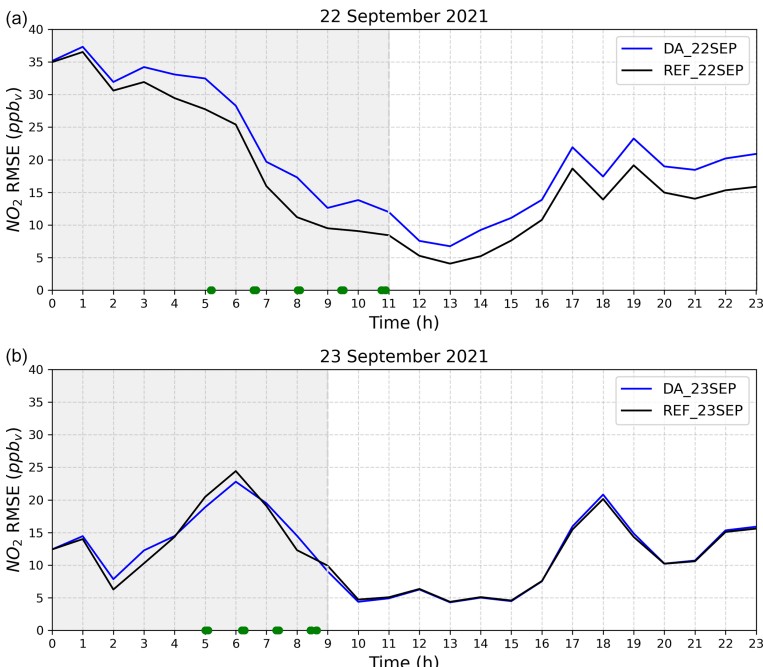

**Figure A2.** Temporal evolution of the RMSE (model − observations) in parts per billion by volume (ppbv) for $NO_2$ calculated for the reference (black) and the analysis (blue) over the 24 h forecast period across all ground stations on 22 September 2021 **(a)** and 23 September 2021 **(b)**. Green dots highlight the time of the assimilated drone profiles. The grey shade illustrates the length of the assimilation window.

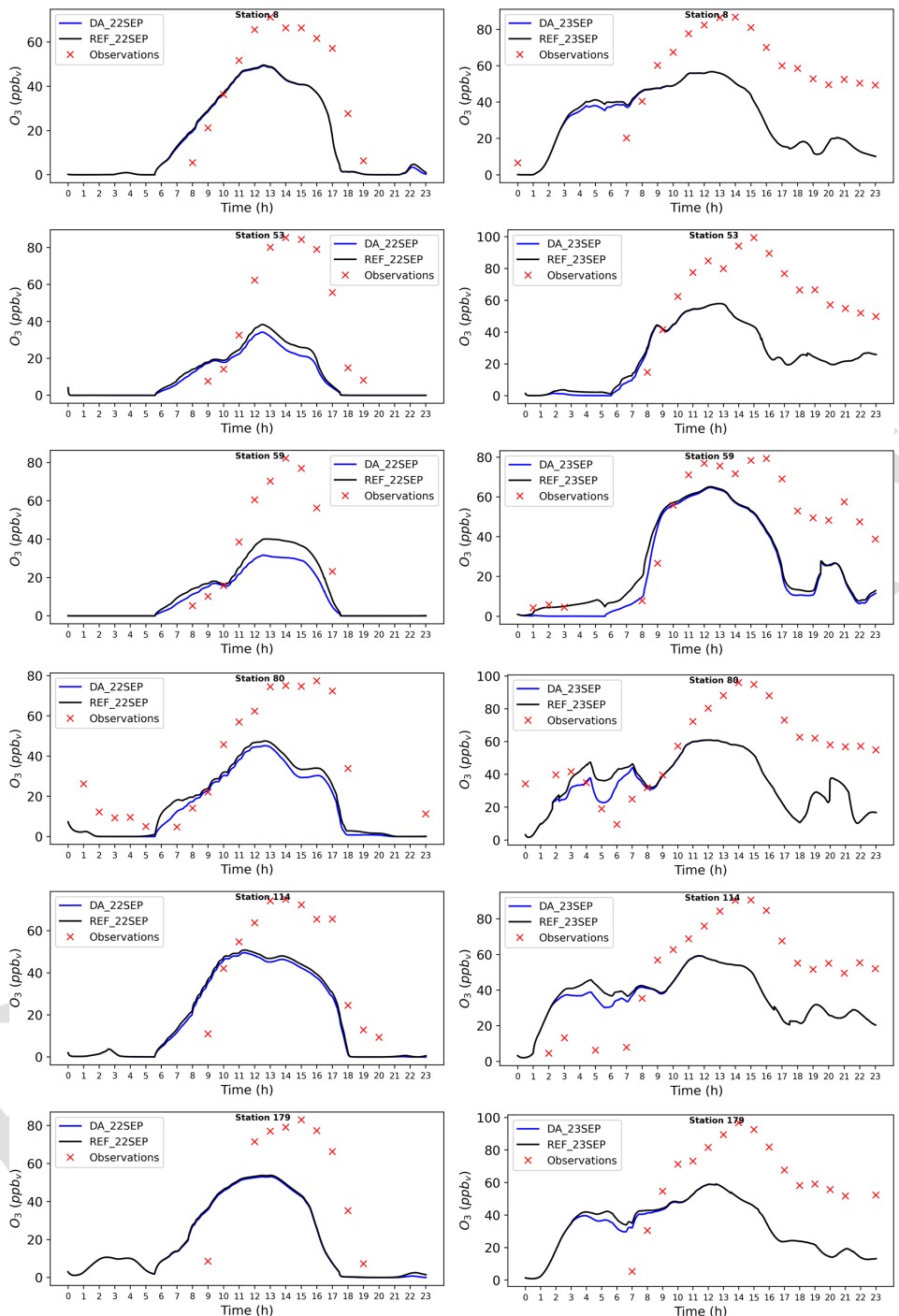

**Figure A3.** Time series of $O_3$ concentrations in parts per billion by volume (ppbv) as measured by ground-based stations and predicted by the model. The left panels show data from 22 September 2021, while the right panels display data from 23 September 2021.

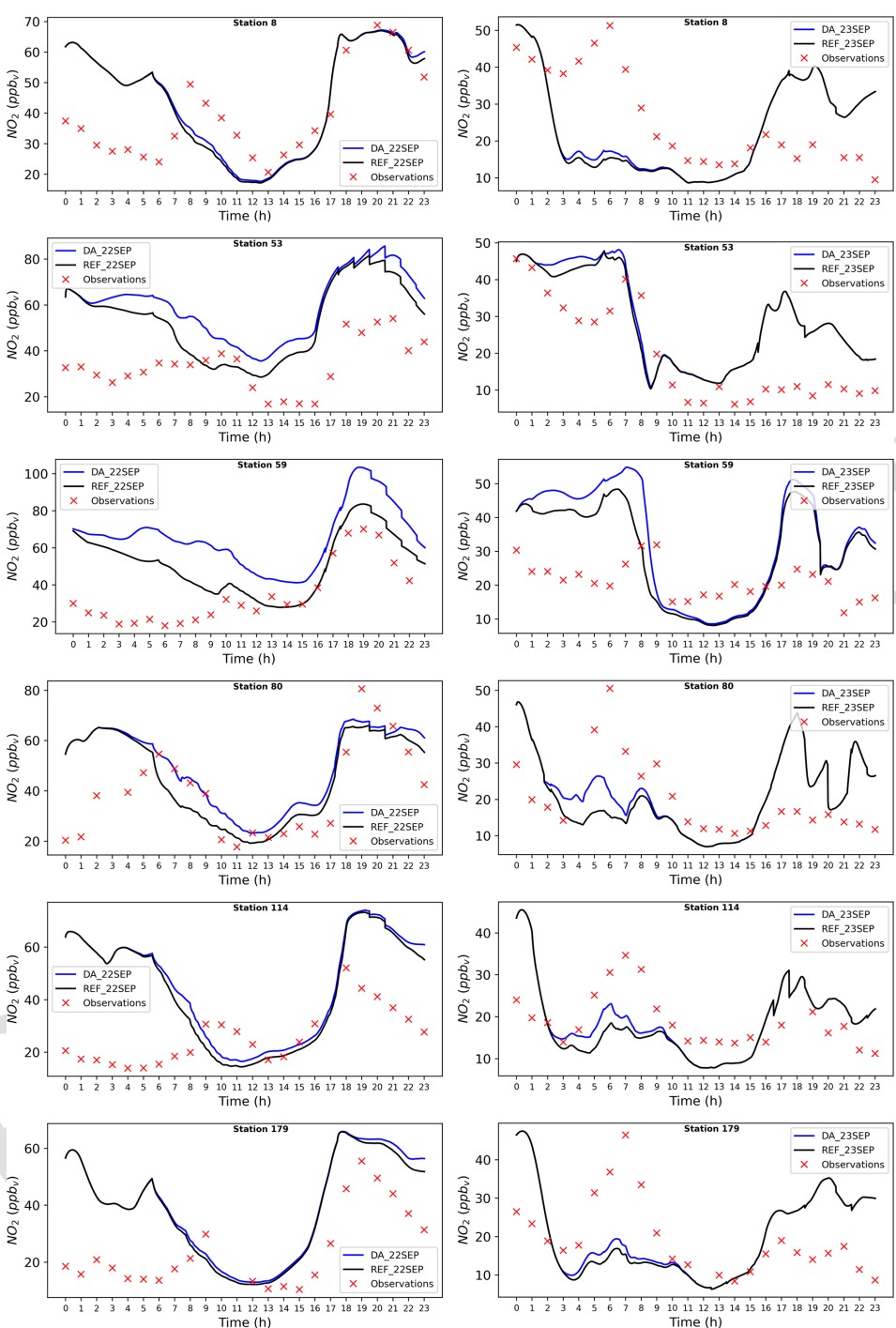

**Figure A4.** Same as Fig. A3 but for $NO_2$.

**Data availability.** The drone data from the MesSBAR campaign used in this study are publicly available from Schlerf et al. (2024) on PANGAEA at the following DOI: https://doi.org/10.1594/PANGAEA.971503.

**Author contributions.** HE and ACL designed the study. HE conducted the simulations and performed the analyses under the scientific supervision of ACL, PF, and AW. TS and RT provided the observational profile data. The manuscript was prepared by HE with the help of all co-authors. All authors reviewed the manuscript.

**Competing interests.** The contact author has declared that none of the authors has any competing interests.

**Disclaimer.** Publisher's note: Copernicus Publications remains neutral with regard to jurisdictional claims made in the text, published maps, institutional affiliations, or any other geographical representation in this paper. While Copernicus Publications makes every effort to include appropriate place names, the final responsibility lies with the authors.

**Acknowledgements.** The authors gratefully acknowledge all the MesSBAR project partners for their valuable efforts in conducting the campaign and processing the data used in this work. We also thank the Federal Highway Research Institute (BASt) for providing the ground-based observations and meteorological data. The authors also gratefully acknowledge the computing time granted through JARA on the supercomputer JURECA (Jülich Supercomputing Centre, 2021) at Forschungszentrum Jülich.

**Financial support.** This research has been supported by the Bundesministerium für Verkehr und Digitale Infrastruktur (mFUND grant no. 19F2097C).

The article processing charges for this open-access publication were covered by the Forschungszentrum Jülich.

**Review statement.** This paper was edited by Kelvin Bates and reviewed by four anonymous referees.

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

## Remarks from the language copy-editor

**CE1**   Firstly, the capital letters are not necessarily the problem here but rather the bold font for the letters, which as per a house standard we avoid except when bold font has a specific use (e.g., highlighting important values in a table). Regarding the capitalisation of acronyms, we try to follow the proper, official name of the model, institution, etc. when possible, but in other cases the default is to follow English rules and only capitalise proper nouns, i.e., not things like numerical weather prediction (NWP). Obviously, this is not always clear as different versions of the name or capitalisation might have been used or published. Please, if you disagree with any of the capitalisation, let me know and we can find a solution. With MesSBAR it is difficult since it is a German abbreviation. I tried to follow the Jülich Forschungszentrum webpage for the English version https://www.fz-juelich.de/en/ice/ice-3/projects/messbar, but is there an official English name for the campaign that can be used? Otherwise I can adjust the capitalisation of "automated airborne measurement of air pollution levels in the near-earth atmosphere in urban areas" to "Automated Airborne Measurement of Air Pollution Levels in the near-Earth Atmosphere in Urban Areas". Please let me know if something else needs to be clarified. Thanks.

## Remarks from the typesetter