# Peer review of "The potential of drone observations to improve air quality predictions by 4D-var"

_EGUsphere, 2024_

## Author Response (AR1)

We would like to thank both reviewers for their thorough work and helpful feedback on our manuscript. The feedback helped to significantly improve the manuscript.
In this document, all remarks by the reviewers are listed in black text, and our corresponding replies are given in blue. In addition to revising the manuscript following the reviewers' comments, we also gave it an editorial review.

**Responses to Anonymous Referee#1**

Drones equipped with low-cost sensors is a new approach for measuring trace gases within the planetary boundary layer. They enable vertical measurements of such gases, complimenting gaps in our current observational system. This manuscript explores a new application of drone observations. The authors use 2-day drone observations of $O_3$ and NO at a polluted site in Germany to perform data assimilation. They found improved performance in simulated levels of $O_3$ and $NO_x$ after assimilation, showing enhancements not only in local vertical profiles but also across a broader spatial region at ground level. This topic is interesting, and the results show some potential for future applications in improving environmental forecasts. The manuscript is well organized and easy to follow.

Thank you very much for your thoughtful review, seeing the potential of our work, and your helpful comments!

My major concern is about the representation error and the correction of emissions. The authors use observations from a single location near the emission sources, to correct a model grid cell averaging 5km x 5km, which introduces representation error.

We agree with referee#1 that a representation error arises due to the mismatch between the spatial scale of the measurement and the spatial scale of the model grid cell used in simulations. However, a representativeness error has been taken into account for our assimilation runs. We included a related discussion in the newest version of the manuscript.
The representation error is calculated considering the model grid cell size $dx$ and the specific characteristics of the observation locations by:

$$\epsilon_{rep} = \sqrt{\frac{dx}{L_x}} \times \epsilon_{abs}.$$

This equation is taken from Elbern et al. (2007), where $L_x$ is the characteristic representativeness length and $\epsilon_{abs}$ is the absolute error.
As suggested, we added the information and assessment of the representativeness error in Section 3.3 (See Lines 170-181).

Indeed, it seems logical for the model to initially underestimate drone-observed NO (due to the model's coarse resolution being unable to resolve fine-scale mobile emissions) and overestimate $O_3$ (because the model's resolution cannot resolve the close-to-source NO titration of $O_3$). In such cases, directly aligning the model with single-spot drone observations and attributing this difference to emissions might result in a significant upward correction of NO emissions, as shown in the authors' results (with correction factors up to 15 on the second day, and the locations of these corrections are suspicious). However, these corrections could be false. Therefore, their findings regarding emission corrections need cautious interpretation and thorough discussion.

Thank you for his comment. We elaborated the discussion of the analysis with respect to the emission correction factors in the manuscript to better explain our results.
Firstly, we noted that the way we presented the correction factors in our analysis could potentially lead to false conclusions. While the emission factors are high for some grid cells, the actual emission rates

are not correspondingly high. In other words, the minimum of the cost function might lead to strong emission correction factors with little impact due to small a priori emissions. To address this, we now present an analysis of the absolute emissions changes by source sector in new Figure 4. This analysis allows us to demonstrate that the emissions on the two days of the study are comparable in terms of the quantity of emissions added and more reasonable when compared to the original emissions.

Secondly, we incorporate a comparison to TROPOMI (TROPOspheric Monitoring Instrument) $NO_2$ tropospheric columns. This satellite data offers valuable insights into the different pollution situations on both days. This explains the discrepancies observed in the emission optimisation results. In addition, We finally discuss the results in light of the wind conditions that mainly determine the location of the emission corrections (Section 4.2). These are different for the analysed days. However, it remains difficult to validate emissions rates in air quality studies. Thus, we refer to the validation of the model results against ground-based observations in order to evaluate the overall impact of the drone data assimilation. This evaluation still allows us to discuss the potential contribution of emissions optimisation to the observed improvements (Section 4.4). Based on these points, we made significant changes, especially in the Sections 4.2 and 4.4 to address your helpful comment.

Section 4.2 has been reworded, and more analysis added.

The old Figure 5 :

[Figure]

Figure 1: NOx emissions from the original emissions inventory (left), the optimised emissions resulting from the data assimilation simulations (middle), and the difference between the original and optimised emissions (right).

The new Figure 4 :

[Figure]

Figure 2: Daily NO$_x$ emissions within the analysed domain (first column) and the analysed NO$_x$ emission changes on 22 September (middle column) and 23 September (last column) 2021. The rows (from top to bottom) display the total NO$_x$ emissions, and the emissions from the public power production, industry, and road transport, respectively.

Other comments are outlined below.

Major comments:

- 1. In Section 2 and 3, could the authors consider adding an introduction on the accuracy and uncertainty of drone observations, and on how the observation error is specifically treated in their 4D-var system?

  We agree that a discussion of the observation errors and uncertainties is of special importance to this study. Following on the referee comments, valuable information regarding the observation error of the drone measurements in the 4D-var system has been incorporated in the manuscript. You will find the major changes in the lines 170-181:

- 2. Line 234-240, what is the authors' definition of "optimal" conditions for efficient emission optimization? How do these conditions affect emission optimization, and are the two cases in

this study considered optimal? Clarifications are needed.

By "optimal wind conditions", we refer to scenarios where the observation site is located on the downwind side of emission sources. In such cases, the observations are directly impacted by the emissions if the diffusion is sufficiently strong to disperse the pollutants effectively. Under these conditions, emissions corrections are more accurate and robust. This is because the wind direction and strength ensure that the observed pollutant concentrations accurately represent the emissions from the sources, as supported by Wu et al. (2020).
Regarding the two cases in this study, the wind conditions differed (see Figure 2 of the manuscript). We have added explanations in Section 4.2, where we discuss the differences between the two days and how they impacted the emissions optimisation results.

- 3. Table 6, what is the definition of "partial cost" and how is it calculated? Could the authors consider adding this information?

Thank you for raising this point. The partial costs refer to the cost term for the initial values $(J_b)$ and for the emissions $(J_e)$, as the 4D-var cost function is composed of $J_b$, the background term, $J_o$, the observation term, and $J_e$, the emission term, with $(J = J_b + J_o + J_e)$. This explanation has been added to line 110. In Table 6, we represent the percentage of these partial costs relative to the total costs $J$ at the analysis iteration. This allows to evaluate if the minimisation algorithm attributed the model-observation discrepancies to uncertainties in the initial values or the emissions.
As suggested, this information has been added to the revised manuscript on lines 340-342 as follows: "..., we present in Table 6 the cost reduction for $O_3$ and NO, as well as the partial costs attributed to the optimisation of the initial values (IV) $\left(\frac{J_b(\mathbf{x_0})}{J(\mathbf{x_0},\mathbf{e})}\right)$ and the emissions correction factors (EF) $\left(\frac{J_e(\mathbf{e})}{J(\mathbf{x_0},\mathbf{e})}\right)$."

- 4. Please change "NOx" to "NO$_x$" throughout the text.

Thanks for spotting these typos. We replaced 'NOx' by 'NO$_x$' in the entire revised manuscript.

We would like to thank both reviewers for their thorough work and helpful feedback on our manuscript. The feedback helped to significantly improve the manuscript.

In this document, all remarks by the reviewers are listed in black text, and our corresponding replies are given in blue. In addition to revising the manuscript following the reviewers' comments, we also gave it an editorial review.

**Responses to Anonymous Referee#2**

Drones bring new opportunities to improve air pollution monitoring vertically within the Planetary Boundary Layer due to their portability, flexibility, and affordability. The authors apply drone profile measurement of $O_3$ and NO to optimize the anthropogenic emissions using the 4D-var data assimilation system of EURAD-IM. As the first application of drone data assimilation within a CTM, research is interesting, which also offers new insights and implications for future studies on emission assimilation. The authors' effort in conducting drone measurements, analyzing the data, and presenting their works are greatly appreciated. However, I still have some concerns in terms of their methodology.

Thank you very much for seeing the value of your work to the community and for the helpful comments you provided. We revised the manuscript significantly taking into account all comments from both referees.

The authors use of drone measurements at a single point to infer NO emissions across a 5 km $\times$ 5 km grid box, which raises concerns regarding data adequacy, potentially introducing significant bias in estimating emission correction factors.

We agree that this setup is not optimal. However at the current stage, the single point profile observations are the only available data set we have for analysing the drones impact on model analyses performing data assimilation. We see the discrepancy regarding the representativeness of the drone data compared to the model resolution. This is nevertheless taken into account by applying a representativeness error to the observation data. This is now comprehensively discussed in the manuscript (see lines 170-181). Moreover, we have extensively worked on the discussion on estimating emission correction factors (Section 4.2). For really proposing emission correction, the statistical evidence is missing. However, the emission corrections are a component of the data assimilation technique being applied and we aim at discussing its full performance.

Particularly, the NO and $NO_2$ emission correction factors derived from DA_23SEP exhibit a 4 to 5-fold increase compared to DA_22SEP. The results seem to be counterintuitive as anthropogenic emissions typically exhibit small changes over two consecutive days, unless extreme events occur that lead to significant changes in NO and $NO_2$ emissions. Even though the authors get 'improved' simulations after the assimilation, I assume the observed large differences in $NO_x$ concentrations would be more related to the daily variations in transport, either horizontally or vertically.

In such cases, improving the representation of the meteorology of the model would likely be more beneficial than merely adjusting pollutant emissions. The authors might consider comparing meteorological parameters in their simulations with observations, particularly focusing on variables like winds, to identify potential discrepancies.

Thank you very much for this comments, which helped us to correct and improve the manuscript. We have identified that the old Figure 5 presenting the change in emissions was misleading. Thanks to your remarks, we decided to replace it with a Figure (Figure 4) illustrating the absolute changes of the daily emission changes. For the correction factors, even if they are large (in some grid boxes), they must be interpreted with caution. The model grid cells with high correction factors are often connected to emission rates that are very low. This is clearly illustrated in the corrected Figure 4 and

Figure A1. In addition, the emissions changes of the two days are comparable and reasonable when compared to the original emissions although different grid cells are affected by the corrections.

The horizontal transport and wind conditions primarily contribute to the observed differences, particularly in terms of location. In fact, during the first day, the wind in the region was low, causing an accumulation of pollutants, while on the second day, the winds were slightly stronger, allowing better transport and dispersion of emissions. We have added to the manuscript a new figure (Fig. 5) representing the tropospheric $NO_2$ column of the TROPOMI satellite, which clearly shows the difference between the pollution situation during the two days. Taking this knowledge into account helps us to explain the difference between the two days in terms of the location and to interpret the amplitude of the emission correction.

The old Figure 5 :

[Figure]

Figure 3: NOx emissions from the original emissions inventory (left), the optimised emissions resulting from the data assimilation simulations (middle), and the difference between the original and optimised emissions (right).

The new Figure 4 :

[Figure]

Figure 4: Daily NO$_x$ emissions within the analysed domain (first column) and the analysed NO$_x$ emission changes on 22 September (middle column) and 23 September (last column) 2021. The rows (from top to bottom) display the total NO$_x$ emissions, and the emissions from the public power production, industry, and road transport, respectively.

The calculated emission correction factors for NO and NO$_2$ can be even larger than 15, suggesting the regional emission inventory they use in their simulation have an uncertainty of over 1400%, which might not be a reasonable value for emission correction. I suggest the authors checking their anthropogenic emission inventory and previous emission assimilation studies to identify a scientifically reasonable range for their emission correction.

We appreciate this comment. We realise that way we presented the correction factor in our initial analysis could potentially lead to false conclusions. Although the emission factors are high for some grid boxes, the absolute emission rates are not correspondingly high and thus did not result an unrealistic emissions change. To address this issue, we have provided an analysis of absolute emissions changes by sector in Figure 4. From previous emission correction studies, we know this behaviour, which appears extreme at the first sight. To propose general emission correction factors to the emission inventory used, it would need a more extensive analysis (more observations for a longer time frame), as the emission input to the chemistry transport model bases on annual emission values per grid cell. The emission corrections analysed here are rather a correction of the time profiles applied to derive

daily emission corrections per grid cell. Section 4.2 has gone through a major transformation to clarify the interpretation of the analysed emission factors.

Additionally, I am not sure whether the authors' observational results would affect by the wind conditions. Both horizontal and vertical wind speeds can exceed the ascent rate (i.e., 1m/s) of their instrument, potentially affecting data collection. How did they keep their instrument ascending at a constant rate in a fixed location? NO is a highly reactive air pollutant which can be converted to $NO_2$ quickly upon emission. Consequently, NO usually presents significant decreasing gradient in the vertical direction. However, the observed NO concentration at 350 m can double the surface NO concentration in the manuscript (e.g., F10, F11). The result is confusing. The authors may want to check the credibility of their data and present relevant explanations.

We understand that a detailed description of the observational setup is missing in our model analysis manuscript. We solely refered to Brettschneider et al. (2022) to find all details about the drone. But we see the necessity to discuss the points addressed in the manuscript. The drone is operated by an autopilot system that uses an inertial navigation solution with an Earth related position based on GNSS data (Global Navigation Satellite System, e.g. GPS, Galileo). During the measurements, the autopilot controls a constant lateral position and a constant vertical climb rate. Wind affects only the attitude of the drone. For low wind situations (like in this study), the effect on the attitude can be neglected, the tilt angle value is low and the assignment of the inflow tube/electrochemical sensors are tilted in wind direction, so the effect on the sensor flow is even less. This information has been added to lines 145-148.
For NO observations, six electrochemical sensors are installed on the drone. The modeling experiment utilizes a dataset from the most efficient sensor. However, all datasets exhibit the same profile shape for flights F10 and F11. This allows us to attribute the higher concentration at altitude, compared to ground level, to actual emissions rather than observational uncertainties. The high NO concentration observed near the ground during the F7/F8/F9 flights is attributed to emissions from the transport sector (Figure 6 shows concentrations exceeding 120 µg/m$^3$ between 4 and 6 UTC). As vertical mixing increases after sunrise, the NO concentration decreases near the ground and rises at higher altitudes. Additionally, emissions from power plants and the industrial sector are released at higher altitudes in the studied area.

I recommend that the authors incorporate more observations, if possible, rather than relying solely on data from a single location, in their assimilation process. By doing so, they can enhance the robustness and credibility of their results, mitigating any potential suspicions regarding the validity of their findings.

We fully agree with your suggestion. Unfortunately, at the moment, we only have data from one measurement campaign available, which was conducted at a single location in September 2021. Because of this, we are presenting our work as a case study to highlight the potential of drone observations for air quality analysis and to encourage more measurement campaigns in the future. Including observational data from other measurement platforms would surely enhance the robustness for example in terms of the emission optimisation, however, would not fit the focus of this manuscript, which is the analysis of the benefits of drone observations. As a future perspective, we hope to collect more observations with such a drone. It is planned to perform similar flights in the Wesseling region on a regular basis for a certain period to enable a statistical evaluation. But also additional campaigns in polluted regions are foreseen.

Specific comments:

- L162: I assume the authors may want to say 'with data assimilation' here or above (i.e., L158)?

Otherwise, all the four experiments are conducted 'without' data assimilation, which is not consistent with what is shown in Table 2.

We are grateful for helping us to avoid the confusion with this phrase. We replaced the sentence: "covering the same period as reference simulations without data assimilation" by "for 22 and 23 September 2021." for a better clarity.

- L218-219 I am not sure whether anthropogenic emissions can have such large differences in two consecutive days.

  Many thanks for raising this point. We fully revised Section 4.2 on the emission optimisation to clarify the aim, the quality, and significance of the analysed emission factors.

- L227-228 As this is a model study, more quantitative analysis is expected.

  Based on your suggestions, we included an additional analysis on the emission corrections (the new Fig. 4), which highlights the sectors subject to significant changes in emissions over the two-day study period. Furthermore, we revised the whole manuscript to enhance the analysis accordingly.

---

## Referee Report (RR1)

**Reviewer Report**

**Manuscript Title:** '*The potential of drone observations to improve air quality predictions by 4D-var*'
**Manuscript Number:** egusphere-2024-517
**Authors:** Hassnae Erraji, Philipp Franke, Astrid Lampert, Tobias Schuldt, Ralf Tillmann, Andreas Wahner, and Anne Caroline Lange

The authors present an interesting and novel study outlining the use of drone-based vertical profile measurements of $O_3$ and NO (collected over a 2-day field campaign) in a 4D-Var data assimilation system coupled to the EURAD-IM CTM to optimize the vertical and spatial distribution several pollutants, as well as the related emission fields. These drones may serve to fill a gap in our observations, particularly within the PBL where few vertically-resolved measurements exist. In general, the manuscript is well written and easy to follow, although I did find there to be quite a lot of minor formatting and grammatical errors throughout.

While the authors do present some reasonable evidence to support the conclusions that the drone observations lead to an improvement in the simulation of these trace-gases and that optimized emissions can be obtained, my largest concern is with the robustness of the results. In particular, I am not entirely convinced with the optimization of the emission fields, given the fact that the assimilation was only performed over a very short time-period and with generally few observations.

**Major Comments:**

- P6, L150-151 – The authors mention they only used the ascent profiles from the drones in the assimilation exercise due to the "higher accuracy". What is exactly meant by this - why were the descent profiles less accurate? Shouldn't it be the same if the ascent/descent rate is fixed at ~1 m/s.

  Similarly, how much does the result change when assimilating both the ascent and descent profiles in the 4D-var system? This could give some indication of how sensitive the assimilation system is to additional measurements (particularly those that might be more uncertain). Due to the overall low number of observations being assimilated over a short window of less than 12 hours on each day, I suspect that this could have a notable influence on the results and is important to investigate.

  The authors even highlight this themselves later on when discussing the optimized emissions; P12, L231-233 "However, their generalization and significance should be rated carefully, mainly because of the limited number of drone profiles being available, the short assimilation windows selected, and the deficiency to perform a long-term statistical analysis".

- From Figure 3, there appears to be a very significant bias in the modeled vs. measured NO profiles on the order of ~30 ppbv, and if this difference was placed in relative terms (in %) it would appear even more extreme. This is only very briefly addressed in Section 4.1, L215: "On both days, the reference simulations underestimate the NO vertical distribution at all heights, with the strongest discrepancies at ground level.". I believe a greater investigation of the source of the bias should be performed or at least explained here, as it may indicate a broader issue in the model (e.g., incorrect NOx chemistry or partitioning between NO and $NO_2$). I think this is particularly crucial if you are going to assume that the partitioning and chemistry of $NO_x$ is correct in the model for the purposes of estimating the $NO_2$ emission corrections based solely on the optimized NO fields.

- P13, L263-264, the authors write "However, it is unfortunately not possible to directly obtain information about the $NO_2$ emissions from the TROPOMI data.". While this is somewhat true, in theory information on the $NO_2$ emissions could be derived from assimilating the S5P observations in the 4D-var system. I'm not necessarily saying this must be done in this paper as it may fall slightly outside of the scope, but it could provide an interesting comparison with those optimized emissions obtained solely by assimilating the drone observations of NO. I think it would also provide increased confidence in the results if separately assimilating both datasets provided a similar result in terms of the optimized emission fields (at least for $NO_2$) for these days.

**Minor Comments:**

- P1, L6: "4D-var takes advantage of the inverse technique…", I think it would be better here to say something along the lines of "4D-var is an inverse modelling technique". There is no single inverse technique, but rather a slew of inverse modelling approaches and methods.
- P2, L25-29, the authors state that very few ground-based monitoring networks exist that provide vertically resolved measurements of these pollutant species. They mention LIDAR and sonde networks, but fail to mention ground-based spectrometer networks such as the Network for the Detection of Atmospheric Composition Change (NDACC) or the Pandonia Global Network (PGN; global network of Pandora UV-Vis spectrometers). These networks provide vertically resolved measurements of $O_3$ and $NO_2$. I suggest that the authors revise this section of the text to include mention of these other networks as well.
- P3, L69: "The aim is to investigate the ability of the 4D-var to adjust". Minor grammatical comment, but I suggest inserting "system" here so it reads "The aim is to investigate the ability of the 4D-var system to adjust".
- General comment on the text on P10 and Table 3 – In the text here, you provide some absolute differences (in ppbv) while later on you start providing relative differences

(in %) for the biases, however in Table 3 you only provide absolute biases (in ppbv). I find this makes it harder to follow. For clarity, I suggest either providing both simultaneously in the text, or choose one and be consistent.

- P12, L226: "The 4d-var data assimilation…" the 'D' should be capitalized here.
- P13, L267-268: "…especially for emissions that are emitted at high altitudes, such as power plants and industries.", I am not really sure the top of a smoke-stack is considered "high altitude" as implied here. Maybe change the text to "higher altitudes", or something similar.
- P14, Figure 4 – In the titles of the left column figures "Nox" should be changed to "$NO_x$". Also, "Septembre" -> "September" in the top titles of the middle and right-most column of figures. I also suggest using a consistent range for the color-bars at least for the middle and right-most column of figures. As it currently stands, upon a quick look it appears as though the road-transport sector has the largest difference, but this is only because the color bar scale is much lower.
- P15, Figure 5 – In the color-bar label, "molec cm-2" should be written as "molec. $cm^{-2}$".
- P15, L285-286, the improvement in the Pearson correlation coefficient of 0.15 here does not coincide with the values listed in Table 4. Only a difference of 0.04 is listed in the table.
- P15, L288-291; the authors state here that a "remarkable improvement" in the $O_3$ concentrations is seen for the beginning of the day on September 23, but neglect to discuss the fact that the optimized simulation agrees more poorly with the observations for the remainder of the day, particularly at the end of the day. I think this should be highlighted more in the text here, as the overall simulation has not necessarily been "improved" with respect to the ground-based observations, only for a short period in which there was originally a large discrepancy. This is discussed to some extent later on P20, but should probably be mentioned here first.
- P16, Table 4; I think it should be made clearer (in the table caption), what values are being provided in the brackets. It does not seem to be mentioned anywhere, are these standard deviations/standard errors?
- General comment on P15-16; similar to my earlier comment about the absolute/relative differences when discussing the comparisons, in the text here on these two pages the authors make statements such as "the assimilation of drone observations results in a strong reduction of the bias by 87% and the RMSE by 20% with an amelioration in the Pearson correlation of 0.15". I believe important context is lost by the authors only providing the differences, but not mentioning in the text what the initial and final values were. For example, if the original bias was 2 ug $m^{-3}$ and the new bias is 1 ug $m^{-3}$, then you can say there was a 50% decrease in the bias, but this would not be as significant as it sounds. I think the text should be amended here to include this.
- P17, Figure 6 – O3 and NO2 in the subplot titles should be subscripted as $O_3$ and $NO_2$.

- P19, Figure 7 – This figure looks a bit too simplistic and like it was made hastily, and I do not think is of publication quality, particularly in relation to the other figures in the manuscript. "O3" in the title should also be subscripted, this also applies to Figure A2, A3, and A4.
- P20, L338-339; the text currently reads "Moreover, the assimilation process allows to obtain optimised emissions rates", this should read "allows *one* to obtain…" or "the assimilation process provides optimized emission rates".
- P20, L348-350 "This is supported by the findings of Wu et al. (2022), affirming that observation at high altitudes can be advantageous for optimising emissions under suitable wind 350 conditions", what exactly is supported by the findings of Wu et al. (2022) here? The authors need to elaborate on this point.
- P20, L361: "Secondly, Some…", "some" should not be capitalized here.
- P20, L362: "(Fig.6, Fig.A3, and Fig.A4)" should have spaces between "Fig." and the figure numbers.
- P21, Figure 8: "NO2" should be subscripted in the y-axis label of the bottom left figure panel.
- P22: The section heading for 'conclusion' should be capitalized.
- P22, L395; Minor comment but Observing System Simulation Experiments (OSSEs) is a more commonly used term for this, not OSE. I am also not quite sure what the authors mean here by " …to assess the advantages and limitations of integrating drone observations into CTMs through the application of a variational data assimilation technique", is that not exactly what was done in this work?

---

## Author Response (AR2)

We would like to sincerely thank both reviewers and the editor for their thorough work and the helpful comments on our manuscript. The feedback helped us to improve the readability, clarify the proceeding of the analysis described and significantly improve the discussion of our manuscript.
In this document, we reply to the reviewers comments as follows: all remarks by the reviewers are listed in black text, and our corresponding replies are given in blue text. In addition to the referee comments, we also edited the copyright statement of Figure 1 directly in the revised manuscript, as proposed by the editorial support.

**Responses to Anonymous Referee#4**

The authors present an interesting and novel study outlining the use of drone-based vertical profile measurements of $O_3$ and NO (collected over a 2-day field campaign) in a 4D-Var data assimilation system coupled to the EURAD-IM CTM to optimize the vertical and spatial distribution several pollutants, as well as the related emission fields. These drones may serve to fill a gap in our observations, particularly within the PBL where few vertically-resolved measurements exist. In general, the manuscript is well written and easy to follow, although I did find there to be quite a lot of minor formatting and grammatical errors throughout.
While the authors do present some reasonable evidence to support the conclusions that the drone observations lead to an improvement in the simulation of these trace-gases and that optimized emissions can be obtained, my largest concern is with the robustness of the results. In particular, I am not entirely convinced with the optimization of the emission fields, given the fact that the assimilation was only performed over a very short time-period and with generally few observations.

We would like to thank Referee#4 very much for seeing the value of our manuscript and for identifying the weaker points of our analysis. We concur that there are a few limitations in the evaluation mainly due to the limited availability of observations. However, we intended to pursue the analysis taking into account the limiting factors and discussing these accordingly. We think that with the analysis, we are still able to highlight the potential of drone observations for atmospheric chemical data assimilation.
In this iteration of the review, we have addressed the remaining critical points raised by both reviewers. We focused on discussing more clearly the limitations imposed by the experimental design and how they should be considered when interpreting the results.

**Major Comments:**

- P6, L150-151–The authors mention they only used the ascent profiles from the drones in the assimilation exercise due to the "higher accuracy". What is exactly meant by this – why were the descent profiles less accurate? Shouldn't it be the same if the ascent/descent rate is fixed at 1m/s.

  Similarly, how much does the result change when assimilating both the ascent and descent profiles in the 4D-var system? This could give some indication of how sensitive the assimilation system is to additional measurements (particularly those that might be more uncertain). Due to the overall low number of observations being assimilated over a short window of less than 12 hours on each day, I suspect that this could have a notable influence on the results and is important to investigate.

  The authors even highlight this themselves later on when discussing the optimized emissions; P12, L231-233 "However, their generalization and significance should be rated carefully, mainly because of the limited number of drone profiles being available, the short assimilation windows selected, and the deficiency to perform a long-term statistical analysis".

We understand the intent of the referee to take the descending profiles into account to include a larger number of profiles. The choice to keep only the ascending profiles is however justified by the design of the drone and the instrumental setup. Since the propellers of the drone are located below the instrumental load, the turbulence created during the descent can disturb the sensor detection. This is in particular evident for the electrochemical sensors even if the ascent/descent rate is constant. By focusing on the ascending profiles only, this interference is minimized due to laminar flow around the sensor surface, such that the data obtained is probing the atmospheric composition more accurately.

Certainly, the number of observations generally impacts assimilation results. However, the measurement error is also a critical factor influencing the analysis. In the case of the drone flight operations only up to an altitude of 350 m, the ascending and descending profiles were taken within a very short temporal window. For most of the flights, the pairs of ascending and descending profiles were thus detected during nearly the same model time step. Thus, assimilating two observations – one more accurate with a smaller observation error, one less accurate with a large observation error – during the same or a subsequent time step would not provide major additional information and not change the results much. Our priority was to retain the most reliable observations to avoid overfitting of the model.

For a better analysis and optimization of emissions, a longer assimilation window with more temporally even distributed observations would be beneficial. For example, additional observations in the afternoon and evening would have provided valuable information about the temporal evolution of pollutants in the lower atmosphere and thus indirectly also information about the occurring emissions.

To clarify the text passages discussed by the referee, we have rephrased them and added the corresponding explanations in the manuscript.

- From Figure 3, there appears to be a very significant bias in the modeled vs. measured NO profiles on the order of ~30 ppbv, and if this difference was placed in relative terms (in %) it would appear even more extreme. This is only very briefly addressed in Section 4.1, L215: "On both days, the reference simulations underestimate the NO vertical distribution at all heights, with the strongest discrepancies at ground level.". I believe a greater investigation of the source of the bias should be performed or at least explained here, as it may indicate a broader issue in the model (e.g., incorrect $NO_x$ chemistry or partitioning between NO and $NO_2$). I think this is particularly crucial if you are going to assume that the partitioning and chemistry of NOx is correct in the model for the purposes of estimating the $NO_2$ emission corrections based solely on the optimized NO fields.

  We would like to express our gratitude for this discussion item. One reason for the discrepancy between the modelled and the observed profiles occurs due to the nature of the model, which relies on relatively static model input parameters (in terms of emissions e.g. not taking into account exceptional emission patterns as for example traffic jams or bypasses due to road closures) that are not designed to reproduce a single "point" observation. This is a well-documented challenge associated with regional chemistry transport models. Local sub-grid emissions, which are not accurately represented in the model, appear to be underestimated in the simulation results as revealed by our analysis.

  A more detailed examination of the origin of the discrepancy is beyond the remit of this manuscript, which explores the potential of drone data to be used for data assimilation. However, we are aware of an imbalance of ozone and nitrogen oxides at night. The $O_3$ minimum is overestimated and $NO_2$ is underestimated (e.g. Lange et al. (2023); Gauss et al. (2024)). This issue is currently under investigation and will be discussed in more detail in a forthcoming manuscript.

- P13, L263-264, the authors write "However, it is unfortunately not possible to directly obtain information about the $NO_2$ emissions from the TROPOMI data.". While this is somewhat true,

in theory information on the $NO_2$ emissions could be derived from assimilating the S5P observations in the 4D-var system. I'm not necessarily saying this must be done in this paper as it may fall slightly outside of the scope, but it could provide an interesting comparison with those optimized emissions obtained solely by assimilating the drone observations of NO. I think it would also provide increased confidence in the results if separately assimilating both datasets provided a similar result in terms of the optimized emission fields (at least for $NO_2$) for these days.

We agree with the referee that assimilating TROPOMI $NO_2$ column data would indeed provide an opportunity to evaluate the analysed emission corrections of our analysis. However, such an additional analysis would be beyond the scope of our manuscript and would introduce a completely different focus as it would include an additional, completely different set of uncertainties. For future 4D-var analyses assimilating drone observations, we can imagine including a comparison with data assimilation analyses assimilating data from ground-based monitoring stations and/or satellite data.

**Minor Comments:**

- P1, L6: "4D-var takes advantage of the inverse technique...", I think it would be better here to say something along the lines of "4D-var is an inverse modelling technique". There is no single inverse technique, but rather a slew of inverse modelling approaches and methods.

  Thank you for bringing this to our attention. We have made the suggested change in the abstract: "4D-var is an inverse modelling technique that allows for simultaneous adjustments of initial values and emissions rates."

- P2, L25-29, the authors state that very few ground-based monitoring networks exist that provide vertically resolved measurements of these pollutant species. They mention LIDAR and sonde networks, but fail to mention ground-based spectrometer networks such as the Network for the Detection of Atmospheric Composition Change (NDACC) or the Pandonia Global Network (PGN; global network of Pandora UV-Vis spectrometers). These networks provide vertically resolved measurements of $O_3$ and $NO_2$. I suggest that the authors revise this section of the text to include mention of these other networks as well.

  We appreciate the referee's comment. Indeed, we overlooked the necessity of mentioning these observation networks. To complete the discussion of vertically resolving networks, the relevant paragraph has been revised and additional information has been incorporated:
  "Similarly, ground-based Fourier Transform InfraRed (FTIR) spectrometers, which from part of the Network for the Detection of Atmospheric Composition Change (NDACC), are capable of retrieving vertically resolved mixing ratios of a range of atmospheric constituents. However, the vertical resolution of these profiles is constrained by their dependence on a priori information, and the network's spatial coverage remains sparse (De Mazière et al., 2018; García et al., 2021). Multi-axis differential optical absorption spectroscopy (MAX-DOAS) is also capable of retrieving trace gas and aerosol vertical profiles (Tirpitz et al., 2021)."

- P3, L69: "The aim is to investigate the ability of the 4D-var to adjust". Minor grammatical comment, but I suggest inserting "system" here so it reads "The aim is to investigate the ability of the 4D-var system to adjust".

  We are grateful to the referee for identifying this error. The recommended correction has been implemented.

- General comment on the text on P10 and Table 3 – In the text here, you provide some absolute differences (in ppbv) while later on you start providing relative differences (in %) for the biases,

however in Table 3 you only provide absolute biases (in ppbv). I find this makes it harder to follow. For clarity, I suggest either providing both simultaneously in the text, or choose one and be consistent.

We thank the referee for pointing out this important remark. To ensure clarity, we have now incorporated both the percentage and the absolute values into the text.

- P12, L226: "The 4d-var data assimilation. . ." the 'D' should be capitalized here.

  Affirmative. This typo has been corrected.

- P13, L267-268: ". . . especially for emissions that are emitted at high altitudes, such as power plants and industries.", I am not really sure the top of a smoke-stack is considered "high altitude" as implied here. Maybe change the text to "higher altitudes", or something similar.

  Thank you for the recommendation. In accordance with your suggestion, the text has been revised to read 'higher altitudes'.

- P14, Figure 4 – In the titles of the left column figures "Nox" should be changed to "$NO_x$". Also, "Septembre" $\rightarrow$ "September" in the top titles of the middle and right-most column of figures. I also suggest using a consistent range for the color-bars at least for the middle and right-most column of figures. As it currently stands, upon a quick look it appears as though the road-transport sector has the largest difference, but this is only because the color bar scale is much lower.

  We would like to thank the referee this pertinent observation. The proposed amendments have been incorporated into the revised Figure 4.

- P15, Figure 5 – In the color-bar label, "molec cm-2" should be written as "molec. $cm^{-2}$".

  We have corrected the label to "molec. $cm^{-2}$" as suggested.

- P15, L285-286, the improvement in the Pearson correlation coefficient of 0.15 here does not coincide with the values listed in Table 4. Only a difference of 0.04 is listed in the table.

  The difference of 0.04 reflects the improvement in the Pearson correlation coefficient during the data assimilation period. In this context, we provide the statistics for the 24-hour forecast, which are listed in parentheses in the Table 3. Thanks to this and a later comment, we have amended Table 4's caption to provide greater clarity on this matter.

- P15, L288-291; the authors state here that a "remarkable improvement" in the $O_3$ concentrations is seen for the beginning of the day on September 23, but neglect to discuss the fact that the optimized simulation agrees more poorly with the observations for the remainder of the day, particularly at the end of the day. I think this should be highlighted more in the text here, as the overall simulation has not necessarily been "improved" with respect to the ground-based observations, only for a short period in which there was originally a large discrepancy. This is discussed to some extent later on P20, but should probably be mentioned here first.

  We agree with the referee's remark. We have amended the wording in aforementioned line to read : "A remarkable improvement in the $O_3$ concentration is noticed within the initial seven hours of the day, while a deterioration is observed between 16:00 and 24:00." As we provide a more detailed explanation later on, we believe this adequately addresses the referee's request for clarification.

- P16, Table 4; I think it should be made clearer (in the table caption), what values are being provided in the brackets. It does not seem to be mentioned anywhere, are these standard deviations/standard errors?

  Thank you for pointing out the lack of clarity in this matter. The values in parentheses represent the statistical data pertaining to the 24-hour forecast. The caption has been revised in order to provide greater clarity. Please find the updated version here: "Statistical comparison of ground-based observations and model outputs (REF: reference run, DA: assimilation run) for $O_3$, NO, and $NO_2$ during the assimilation window and, in parentheses, the 24-hour forecast on 22-23 September 2021. The Bias and RMSE are in $\mu$gm$^{-3}$."

- General comment on P15-16; similar to my earlier comment about the absolute/relative differences when discussing the comparisons, in the text here on these two pages the authors make statements such as "the assimilation of drone observations results in a strong reduction of the bias by 87% and the RMSE by 20% with an amelioration in the Pearson correlation of 0.15". I believe important context is lost by the authors only providing the differences, but not mentioning in the text what the initial and final values were. For example, if the original bias was 2 $ug$ $m^{-3}$ and the new bias is 1 $ug$ $m^{-3}$, then you can say there was a 50% decrease in the bias, but this would not be as significant as it sounds. I think the text should be amended here to include this.

  We thank the referee for these pertinent remarks, which have been addressed in the revised manuscript. In the revised version, we now provide both percentages the corresponding absolute values.

- P17, Figure 6 – O3 and NO2 in the subplot titles should be subscripted as $O_3$ and $NO_2$.

  $O_3$ and $NO_2$ are now correctly subscripted in the Figure 6.

- P19, Figure 7 – This figure looks a bit too simplistic and like it was made hastily, and I do not think is of publication quality, particularly in relation to the other figures in the manuscript. "O3"in the title should also be subscripted, this also applies to Figure A2, A3, and A4.

  Indeed. The figures have been revised in order to enhance their visual quality and ensure alignment with the established publication standards. All notation errors have now been corrected across Figures 7, A2, A3, and A4.

- P20, L338-339; the text currently reads "Moreover, the assimilation process allows to obtain optimised emissions rates", this should read "allows *one* to obtain..."or "the assimilation process provides optimized emission rates".

  We thank the referee for pointing out this wording mistake. The phrase has been corrected as follows: "Moreover, the assimilation process provides optimised emissions rates for each day."

- P20, L348-350 "This is supported by the findings of Wu et al. (2022), affirming that observation at high altitudes can be advantageous for optimising emissions under suitable wind 350 conditions", what exactly is supported by the findings of Wu et al. (2022) here? The authors need to elaborate on this point.

  Thanks for identifying the lack of information. For better clarity, this sentence has been edited as follows: "In a recent study by Wu et al. (2022), it was demonstrated that for high-altitude observations, the efficiency of emission rate optimization is conditioned by favorable wind conditions and strong vertical diffusion."

- P20, L361: "Secondly, Some...", "some" should not be capitalized here.

  This typo has been corrected in the revised manuscript.

- P20, L362: "(Fig.6, Fig.A3, and Fig.A4)" should have spaces between "Fig." and the figure numbers.

  Affirmative. This mistake has been corrected in the revised manuscript.

- P21, Figure 8: "NO2" should be subscripted in the y-axis label of the bottom left figure panel.

  Sure. This has been corrected in the revised manuscript.

- P22: The section heading for 'conclusion' should be capitalized.

  Thanks. This typo has been corrected in the revised manuscript.

- P22, L395; Minor comment but Observing System Simulation Experiments (OSSEs) is a more commonly used term for this, not OSE. I am also not quite sure what the authors mean here by "...to assess the advantages and limitations of integrating drone observations into CTMs through the application of a variational data assimilation technique", is that not exactly what was done in this work?

  We thank the referee for highlighting this important point. The sentence was indeed poorly formulated. Our future work aims to conduct OSSEs to assess the added value of assimilating drone observations into operational systems, alongside conventional observations type. The primary objective is to determine whether integrating drone data with conventional observations can further improve the assessment of local emissions and refine the vertical distribution of pollutants in the model. This was not feasible in the current work due to limited observation data from the campaign. Positive results from OSSEs might advocate for a wider integration of drone observations in operational forecasting systems.
  The term OSSEs has been corrected and the related phrase has been modified as follows: "From a future perspective, a valuable extension of this work will be to conduct Observing System Simulation Experiments (OSSEs) to evaluate the added value of integrating drone-based observations into the air quality forecasting system, compared to conventional observations such as ground-based measurements and satellite data."

**Responses to Anonymous Referee#3**

This manuscript describes a modeling study in which the emissions of mainly nitrogen oxides are adjusted by assimilating drone-based observations of ozone and nitrogen monoxide. This is a interesting topic, since understanding how to use drone observations in chemical data assimilation could open up new pathways to quantify local emission sources. Overall, I find the study technically solid, and as it stands, I have no strong objections to its publication. However, the paper would become more interesting if a more stronger analysis of the mechanisms taking place in the inversion could be included. The current discussion is detailed enough, but not very conclusive.

We appreciate the review comments Referee#3 and their recognition of the significance of our study. During this review process, we have incorporated several amendments to the manuscript with the objective of strengthening the discussion on the potential of drone observations for data assimilation in the context of trace gas forecasts. Please find our responses to the more detailed points below.

For example, why did the model fail to capture the $O_3$ levels on the night of 23rd September, why did

the assimilation system resolve the discrepancy by a large emission adjustment (as opposed to initial values), and to what extent are the optimized emissions supported or explained by independent data? At minimum, it could be useful to examine the spatial scale of the patterns that the assimilation aims to capture. Figs. A3 and A4 show quite high heterogeneity in both $O_3$ and $NO_2$ on the night of 23rd, which suggests that the assimilation result might be sensitive to model resolution. On the other hand, if the patterns are region-wide, they might be rather driven by a model rather than emission inventory bias.

We agree with the referee that these are valid and important questions given the direct comparison of model results and observations. Regarding the first question, we agree that the model does not well predict the $O_3$ and $NO_2$ concentrations during the night of 23rd September. These results can be attributed to both the model and the emissions. On the model side, one potential factor we suspect is the representation of the planetary boundary layer height, which could significantly affect nighttime predictions. The issue is known and currently under investigation. Since it is independent of the assimilation process, we considered a comprehensive evaluation beyond the scope of our manuscript. However we approach the emissions that are one of the largest sources of uncertainty in model predictions. The objective of utilising the 4D-Var method is to improve the representation of emissions, and thereby enhancing the overall model forecast. In our results, we observed a limited and minor improvement during the night, which we attribute to the fixed temporal profile used for the emissions optimisation. This topic has been discussed in detail in section 4.3.1. Another reason can be the lack of observability – preferable information content – provided by the limited number of drone profiles. In response to the second question, the greater impact of optimizing emissions compared to initial values is primarily due to the inversion process being more effective under favorable/higher wind conditions. Finally, it is challenging to directly validate the optimized emissions with observational data. However, the validation of the analysis against ground-based observations suggests that the optimized emissions lead to a superior analysis, showing that the optimization is moving in the desired direction. This indicates that the uncertainty in the a priori emission values has been reduced, despite the limitation imposed by the short data assimilation window.

Related to the model resolution, it might be worth noting that the representation errors of a single drone profile might not be independent of each other. This would violate the assumption of a diagonal R matrix and effectively result in overstating the accuracy of the assimilated measurements.

We thank the referee for pointing out this important remark. We agree that both measurement and representation errors in the observations are likely correlated. However, the unavailability of detailed error estimates makes it challenging to accurately quantify these correlations. Therefore, in our simulations, we assume that the errors are uncorrelated. To compensate for the absence of correlation, we have increased the error variances to their maximum reasonable values, thereby ensuring that the observations are given appropriate weight in the assimilation.
The topic of the representation error is of significant interest and worthy of discussion. The work of Janjić et al. (2018) provides a detailed examination of the various definitions of this error and current research is exploring an methodology for assessing this error in EURAD-IM, with the view to conducting similar analysis.

**Minor Comments:**

- L120: Does the block-wise structure of K relate to chemical or spatial correlations, or both?

    The matrix K accounts for both chemical and spatial correlations. To enhance the clarity of the sentence, we have modified it in the manuscript as follows: "The matrix **K** is defined as block diagonal, with non-zero entries for correlations between species and near-by emissions. The variance and correlation values are provided in Paschalidi (2015)."

- Section 3.1: What was the vertical resolution of the profiles? How many data points did the assimilated profiles consist of?

  The vertical resolution of these profiles is approximately 10 meters, with in total 254 data points assimilated on 22 September 2021 and 257 on 23 September 2021 for both $O_3$ and NO. We added this information to the manuscript.

- L190: Would the discussion of background error correlation lengths fit better to Section 2.2?

  We see the referee's point of discussing the background error correlation length in the section, where we introduce the data assimilation method. However, since we treat the correlation length differently for different analysis setups, we decided and now reassure that it is better to be discussed in the simulations setup section, as it is for example model grid resolution dependent. Therefore, we prefer to leave the statement where it is placed now.

- L265: How much is 16 $Mgd^{-1}$ relative to the total daily emission over the region in Fig. 1?

  The emission of 16 $Mgd^{-1}$ represents approximately 3.46% of the total daily $NO_x$ emissions in the analyzed region, where the total daily $NO_x$ emission is about 462 $Mgd^{-1}$.. This information has been added into the manuscript.

- L363: "constrains the optimisation to more flexible adjustments..." - not sure if I understand.

  Thank you for pointing out this confusing statement. We rephrased it to: "It is assumed that the temporal emission profile is more certain than the emission strength. Deriving e.g. hourly emission factors instead would allow for more flexible adjustments of the emissions, which would be beneficial for the nowadays strongly regulated emission sources, such as the power production (dependent on the availability of renewable energy)."

- Fig. 1: N and E seem to be swapped in the tick labels

  We appreciate the attention of the referee. This mistake has been corrected in the revised manuscript.

- Fig. 8: The vertically oriented row labels (DA_23SEP – REF_23SEP etc.) are small and difficult to read; please consider showing at least the chemical species more clearly.

  We thank the referee for this recommendation. The figure has been improved in the revised manuscript.

**Bibliography**

De Mazière, M., Thompson, A. M., Kurylo, M. J., Wild, J. D., Bernhard, G., Blumenstock, T., Braathen, G. O., Hannigan, J. W., Lambert, J.-C., Leblanc, T., McGee, T. J., Nedoluha, G., Petropavlovskikh, I., Seckmeyer, G., Simon, P. C., Steinbrecht, W., and Strahan, S. E.: The Network for the Detection of Atmospheric Composition Change (NDACC): history, status and perspectives, Atmospheric Chemistry and Physics, 18, 4935–4964, https://doi.org/10.5194/acp-18-4935-2018, 2018.

García, O. E., Schneider, M., Sepúlveda, E., Hase, F., Blumenstock, T., Cuevas, E., Ramos, R., Gross, J., Barthlott, S., Röhling, A. N., Sanromá, E., González, Y., Gómez-Peláez, A. J., Navarro-Comas, M., Puentedura, O., Yela, M., Redondas, A., Carreño, V., León-Luis, S. F., Reyes, E., García, R. D., Rivas, P. P., Romero-Campos, P. M., Torres, C., Prats, N., Hernández, M., and López, C.: Twenty years of ground-based NDACC FTIR spectrometry at Izaňa Observatory – overview and long-term comparison to other techniques, Atmospheric Chemistry and Physics, 21, 15 519–15 554, https://doi.org/10.5194/acp-21-15519-2021, 2021.

Gauss, M., Petiot, V., Joly, M., Besson, F., Royer, A., Douros, J., Tsikerdekis, A., Eskes, H. J., Bennouna, Y., Thouret, V., Friese, E., and Lange, A. C.: Quarterly report on the evaluation of EURAD-IM NRT productions (daily analyses and forecasts) March 2024 - April 2024 - May 2024, Evaluation report, Norwegian Meteorological Institute, https://atmosphere.copernicus.eu/sites/default/files/custom-uploads/EQC-regional/MAM-2024/CAMS283_2021SC2_D83.1.4.1-2024Q2_202407_EURAD-IM_EQC_Report_v1.pdf, 2024.

Janjić, T., Bormann, N., Bocquet, M., Carton, J. A., Cohn, S. E., Dance, S. L., Losa, S. N., Nichols, N. K., Potthast, R., Waller, J. A., and Weston, P.: On the representation error in data assimilation, Quarterly Journal of the Royal Meteorological Society, 144, 1257–1278, https://doi.org/https://doi.org/10.1002/qj.3130, 2018.

Lange, A. C., Franke, P., Backes, P., and Elbern, H.: Immissionsseitige Bewertung der Luftschadstoff-Emissionen einzelner Quellen und Anpassung der nationalen Emissionsdaten zur Beurteilung der Luftqualität, Project report 149/2023, Umweltbundesamt, https://www.umweltbundesamt.de/publikationen/immissionsseitige-bewertung-der-luftschadstoff, 2023.

Paschalidi, Z.: Inverse Modelling for Tropospheric Chemical State Estimation by 4-Dimensional Variational Data Assimilation from Routinely and Campaign Platforms, Ph.D. thesis, University of Cologne, 2015.

Tirpitz, J.-L., Frieß, U., Hendrick, F., Alberti, C., Allaart, M., Apituley, A., Bais, A., Beirle, S., Berkhout, S., Bognar, K., Bösch, T., Bruchkouski, I., Cede, A., Chan, K. L., den Hoed, M., Donner, S., Drosoglou, T., Fayt, C., Friedrich, M. M., Frumau, A., Gast, L., Gielen, C., Gomez-Martín, L., Hao, N., Hensen, A., Henzing, B., Hermans, C., Jin, J., Kreher, K., Kuhn, J., Lampel, J., Li, A., Liu, C., Liu, H., Ma, J., Merlaud, A., Peters, E., Pinardi, G., Piters, A., Platt, U., Puentedura, O., Richter, A., Schmitt, S., Spinei, E., Stein Zweers, D., Strong, K., Swart, D., Tack, F., Tiefengraber,

M., van der Hoff, R., van Roozendael, M., Vlemmix, T., Vonk, J., Wagner, T., Wang, Y., Wang, Z., Wenig, M., Wiegner, M., Wittrock, F., Xie, P., Xing, C., Xu, J., Yela, M., Zhang, C., and Zhao, X.: Intercomparison of MAX-DOAS vertical profile retrieval algorithms: studies on field data from the CINDI-2 campaign, Atmospheric Measurement Techniques, 14, 1–35, https://doi.org/10.5194/amt-14-1-2021, 2021.